# Immunoglobulin M regulates airway hyperresponsiveness independent of T helper 2 allergic inflammation

Sabelo Hadebe[1]*, Anca Flavia Savulescu[2], Jermaine Khumalo[1], Katelyn Jones[1], Sandisiwe Mangali[1,3], Nontobeko Mthembu[1], Fungai Musaigwa[1], Welcome Maepa[2], Hlumani Ndlovu[2], Amkele Ngomti[1], Martyna Scibiorek[1,3], Javan Okendo[2,4], Frank Brombacher[1,3,5]*

[1]Division of Immunology, Department of Pathology, Faculty of Health Sciences, University of Cape Town, Cape Town, South Africa; [2]Division of Chemical, Systems & Synthetic Biology, Faculty of Health Sciences, Institute of Infectious Disease & Molecular Medicine, University of Cape Town, Cape Town, South Africa; [3]International Centre for Genetic Engineering and Biotechnology (ICGEB) and Institute of Infectious Diseases and Molecular Medicine (IDM), Division of Immunology, Health Science Faculty, University of Cape Town, Cape Town, South Africa; [4]Centre for Research in Therapeutic Sciences (CREATES), Strathmore University, Nairobi, Kenya; [5]Wellcome Centre for Infectious Diseases Research in Africa (CIDRI-Africa), Institute of Infectious Diseases and Molecular Medicine (IDM), Faculty of Health Sciences, University of Cape Town, Cape Town, South Africa

*For correspondence:
sabelo.hadebe@uct.ac.za (SH);
brombacherfrank@gmail.com
(FB)

**Competing interest:** The authors declare that no competing interests exist.

## eLife Assessment

This **important** study demonstrates a reduction in airway hyperresponsiveness (one of the mechanisms of allergic asthma) in the absence of IgM in a house dust mite-induced mouse model of allergic asthma. While this result suggests a new mechanistic role for IgM, the proposed new function is not as yet robustly supported by the current experiments and thus the evidence remains **incomplete**. A connection between the findings and human disease is not established so far, but the study will be interest to clinical immunologists.

**Abstract** Allergic asthma is a disease driven by T helper 2 (Th2) cells, eosinophilia, airway hyperresponsiveness (AHR), and IgE-secreting B cells. Asthma is largely controlled by corticosteroids and $\beta_2$ adrenergic receptor agonists that target and relax airway smooth muscle (ASM). Immunoglobulin M (IgM) isotype secreted by naïve B cells is important for class switching but may have other undefined functions. We investigated the role of IgM in a house dust mite (HDM)-induced Th2 allergic asthma model. We sensitised wild-type (WT) and IgM-deficient (IgM KO) mice with HDM and measured AHR, and Th2 responses. We performed RNA sequencing on the whole lung of WT and IgM KO mice sensitised to saline or HDM. We validated our AHR data on human ASM by deleting genes using CRISPR and measuring contraction by single-cell force cytometry. We found IgM to be essential in AHR but not Th2 airway inflammation or eosinophilia. RNA sequencing of lung tissue suggested that IgM regulated AHR through modulating brain-specific angiogenesis inhibitor 1-associated protein 2-like protein 1 (*Baiap2l1*) and other genes. Deletion of *BAIAP2L1* led to a differential reduction in human ASM contraction when stimulated with TNF-α and Acetylcholine, but not IL-13. These findings have implications for future treatment of asthma beyond current therapies.

## Introduction

Immunoglobulin M (IgM) is the first antibody isotype expressed during B cell development and the first humoral antibody responder, conserved across all species from zebrafish to humans (*Akula et al., 2014*). IgM can be divided into natural and antigen-induced IgM and can either be membrane-bound IgM-type B cell receptor (BCR) or secreted IgM (*Baumgarth et al., 2000*; *Blandino and Baumgarth, 2019*). Natural IgM plays multiple roles in homeostasis, including scavenging and clearance of apoptotic cell debris in conjunction with phagocytic macrophages, B cell survival through tonic signals, lymphoid tissue architecture, and prevention of autoimmune diseases (*Ehrenstein and Notley, 2010*; *Michaud et al., 2020*; *Quartier et al., 2005*). At mucosal sites, both natural and antigen-induced IgM play a role in shaping healthy microbiota, and IgM repertoire is also shaped by microbiota (*New et al., 2020*; *Wesemann et al., 2013*). Secreted IgM antigen complexes can connect signals via unique and shared receptors, suggestive of a more pleiotropic role in homeostasis and disease states (*Jones et al., 2020*; *Kawahara et al., 2003*; *Nguyen et al., 2017a*).

Natural IgM and secreted IgM are essential in many infectious and non-infectious diseases, including those induced by parasites, fungi, bacterial, viral, and autoimmune diseases (*Jones et al., 2020*). In *Plasmodium falciparum*, anti-α-gal IgM and anti-MSP1 directed antibodies are protective against primary and secondary infections (*Krishnamurty et al., 2016*; *Yilmaz et al., 2014*). Mice deficient in secreted IgM are susceptible to pulmonary *Cryptococcus neoformans* and *P. carinii* infection partly due to reduced activation of innate and adaptive responses (*Rapaka et al., 2010*; *Subramaniam et al., 2010*). At mucosal surfaces, IgM promotes healthy gut bacteria that is beneficial for homeostasis, such as Firmicutes and Bacteroidetes (*Magri et al., 2017*). Natural and induced secreted IgM produced mainly by B1a cells is protective against *Streptococcus pneumoniae* and *Francisella tularensis* infection (*del Barrio et al., 2015*; *Weber et al., 2014*). In these settings, the protective effects of sIgM depended on cytokines IL-1β and GM-CSF (*del Barrio et al., 2015*; *Weber et al., 2014*). The involvement of natural or induced IgM in allergic asthma is unknown, despite selective IgM syndrome is dominated by asthma patients (*Goldstein et al., 2006*).

Asthma is a T helper 2 (Th2) disease characterised by eosinophilic lung inflammation, mucus production, AHR, Th2 cytokines (interleukin-4 (IL-4), IL-5, and IL-13), and B cells producing IgE (*Lambrecht and Hammad, 2015*). IgM is central to class switch recombination that results in IgE class-switched B cells (*Mandler et al., 1993*). We and others recently showed that the role of B cells in asthma is complex, where the load of the antigen is crucial in their function (*Dullaers et al., 2017*; *Habener et al., 2021*; *Hadebe et al., 2021*). Mice deficient of B cells (μMT KO) can mount exaggerated AHR when challenged with HDM (*Ballesteros-Tato et al., 2016*; *Dullaers et al., 2017*; *Wypych et al., 2018*) or ovalbumin (OVA) (*Hamelmann et al., 1999*; *Korsgren et al., 1997*; *MacLean et al., 1999*) partly due to lack of regulatory B cells that dampen AHR (*Habener et al., 2021*). Furthermore, interleukin 4 receptor alpha signalling in B cells is required for optimal Th2 allergic airway inflammation through regulation of AHR, germinal centre (GC) formation, and B effector 2 function (*Hadebe et al., 2021*).

Interestingly, B cell isotypes show unique functions in allergic asthma, for example, IgE and its high-affinity receptor, FcεR are redundant in an HDM model (*McKnight et al., 2017*), whereas IgD plays an amplifying and regulatory role in various allergic models (*Shan et al., 2018*). In this context, IgD activates basophils to secrete IL-4 and Th2 induction during the sensitisation stage through binding basophils via galectin-9 and CD44. Once Th2 responses have been amplified, IgD ligation blocked IgE-mediated basophil degranulation through competing for antigen and inhibiting FcεR-mediated signalling (*Shan et al., 2018*). How IgM isotype contributes in the development of allergic asthma is unclear. It is also unclear whether secreted IgM plays different roles compared to membrane-bound IgM, which is more likely to undergo class switching to IgE. We challenged mice lacking both membrane and secreted IgM with HDM and other allergens and found a profound reduction in AHR. RNA sequencing of lung tissue showed a downregulation of brain-specific angiogenesis inhibitor 1-associated protein 2-like protein 1 (*Baiap2l1*) and erythroid differentiation regulatory factor 1 (*Erdr1*) genes associated with actin cytoskeleton and rearrangement smooth muscle contraction. As a proof of principle, we showed in human smooth muscle cell line that deletion of these genes via CRISPR resulted in a reduction in smooth muscle contraction at a single cell level. These are unexpected functions of secreted and membrane-bound IgM; namely, its involvement in modulating airway smooth muscle contraction.

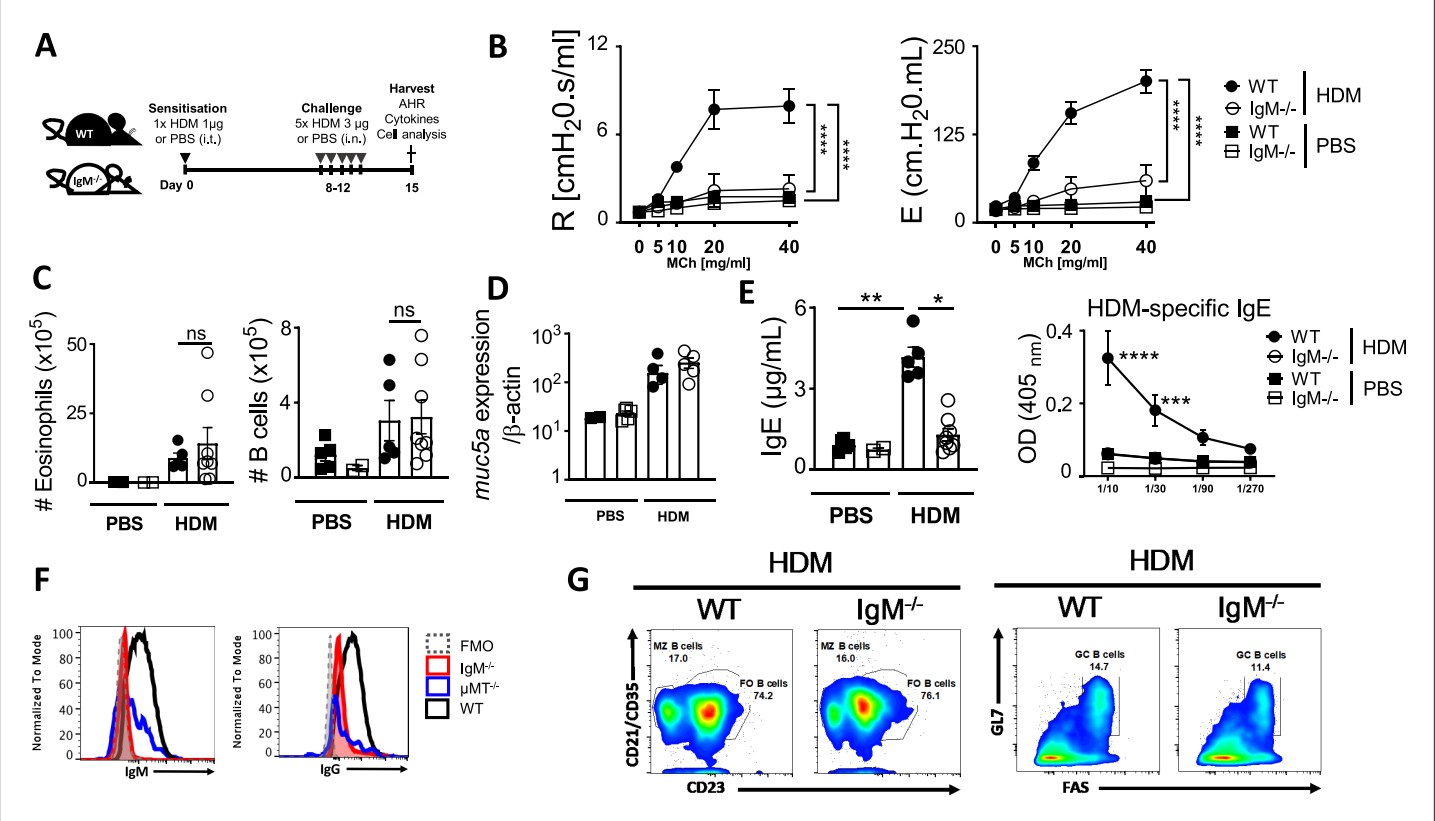

**Figure 1.** IgM deficiency leads to reduced airway hyperresponsiveness and class switching to Immunoglobulin E (IgE) in house dust mite (HDM)-induced asthma. (**A**) Schematic diagram showing sensitisation and challenge protocol where mice (IgM KO) and wild-type littermate control (wild-type, WT) were sensitised with HDM 1 μg intra-tracheally on days 0 and challenged with HDM 3 μg on days 8–12. Analysis was done on day 15. (**B**) Airway resistance and elastance were measured with increasing doses of acetyl methacholine (0–40 mg/mL). (**C**) Total lung eosinophil numbers (live+Siglec-F+CD11c-) and B cells (live+B220+CD19+MHCII+) were stained and analysed by flow cytometry and enumerated from % of live cells. (**D**) *Muc5a* gene expression in whole lung tissue. (**E**) Total IgE and HDM-specific IgE in serum. (**F**) IgG1 and IgM surface expression in mediastinal lymph node B cells of WT, IgM KO, and μMT KO mice. (**G**) Marginal Zone (live+B220+CD19+MHCII+CD21/CD35+CD23-), follicular (live+B220+CD19+MHCII+CD23+CD21/CD35+) and Germinal Centre (live+B220+CD19+MHCII+GL7+FAS+) B cells in the mediastinal lymph node of WT and IgM KO mice challenged with HDM. Shown is mean ± SEM from two pooled experiments (n=7–10). Significant differences between groups were performed by Student *t*-test (Mann-Whitney) (**c, d, e**) or by two-way ANOVA with Benferroni post-test (**b**) and are described as: *$p<0.05$, **$p<0.01$, ***$p<0.001$, ****$p<0.0001$.

The online version of this article includes the following figure supplement(s) for figure 1:

**Figure supplement 1.** Immunoglobulin M (IgM) deficiency leads to reduced airway hyperresponsiveness, but eosinophils and B cells are unaffected.

**Figure supplement 2.** Reduced airway hyperresponsiveness in Immunoglobulin M (IgM)-deficient mice is independent of allergen.

**Figure supplement 3.** Reduced airway hyperresponsiveness in Immunoglobulin M (IgM)-deficient mice is independent of mouse background in house dust mite (HDM)-induced asthma.

**Figure supplement 4.** B cell development is not impacted by lack of Immunoglobulin M (IgM); however, IgD expression is upregulated in all tissues.

## Results

### IgM-deficient mice show profound airway hyperresponsiveness reduction when exposed to HDM

We sensitised and challenged IgM-deficient (IgM KO) and wild-type Balb/c (WT) mice with HDM intratracheally (i.t.) and analysed AHR, lung infiltrates, and cytokines (*Figure 1A*, *Figure 1—figure supplement 1A*). We found moderately reduced resistance and elastance in IgM KO sensitised with a high dose of HDM (100 μg) compared to WT mice (*Figure 1—figure supplement 1B*). Similarly, we also observed a profound reduction in AHR in IgM KO mice sensitised with a low dose of HDM (1 μg) compared to WT (*Figure 1B*). Interestingly, lung eosinophils were intact at both low dose (*Figure 1C*) and high dose HDM (*Figure 1—figure supplement 1D*). We could also show that AHR reduction in IgM KO mice was not allergen-specific, as we observed similar findings using ovalbumin (OVA)

complexed to alum adjuvant (*Figure 1—figure supplement 2A–B*) and acute papain-induced allergic inflammation (*Figure 1—figure supplement 2C–D*). We observed similar findings of profound AHR reduction in IgM KO mice in C57BL/6 background compared to WT mice (*Figure 1—figure supplement 3A*). These findings suggested that IgM-deficient mice have profound AHR reduction independent of allergen or mice background strain.

## IgM deficiency does not impact B cell subsets in primary and secondary lymphoid organs, but class switching is impaired

We also observed no significant changes between IgM KO and WT mice in mucus production shown by *Muc5a* gene expression (*Figure 1D*) or goblet cells in Balb/C (*Figure 1—figure supplement 1F*) and C57BL/6 mice challenged with HDM (*Figure 1—figure supplement 3B*). Accompanying reduction in AHR in IgM KO mice was low titers of total IgE, HDM-specific IgE, and mediastinal lymph node (mLN) B cell surface expression of IgM and IgG1 (*Figure 1E–F*), owing to a lack of class switching. Interestingly, the numbers of lung B cells were normal (*Figure 1C*, *Figure 1—figure supplement 1C*) and the frequencies of B cell subsets, such as follicular, marginal zones, and germinal centres (GCs) B

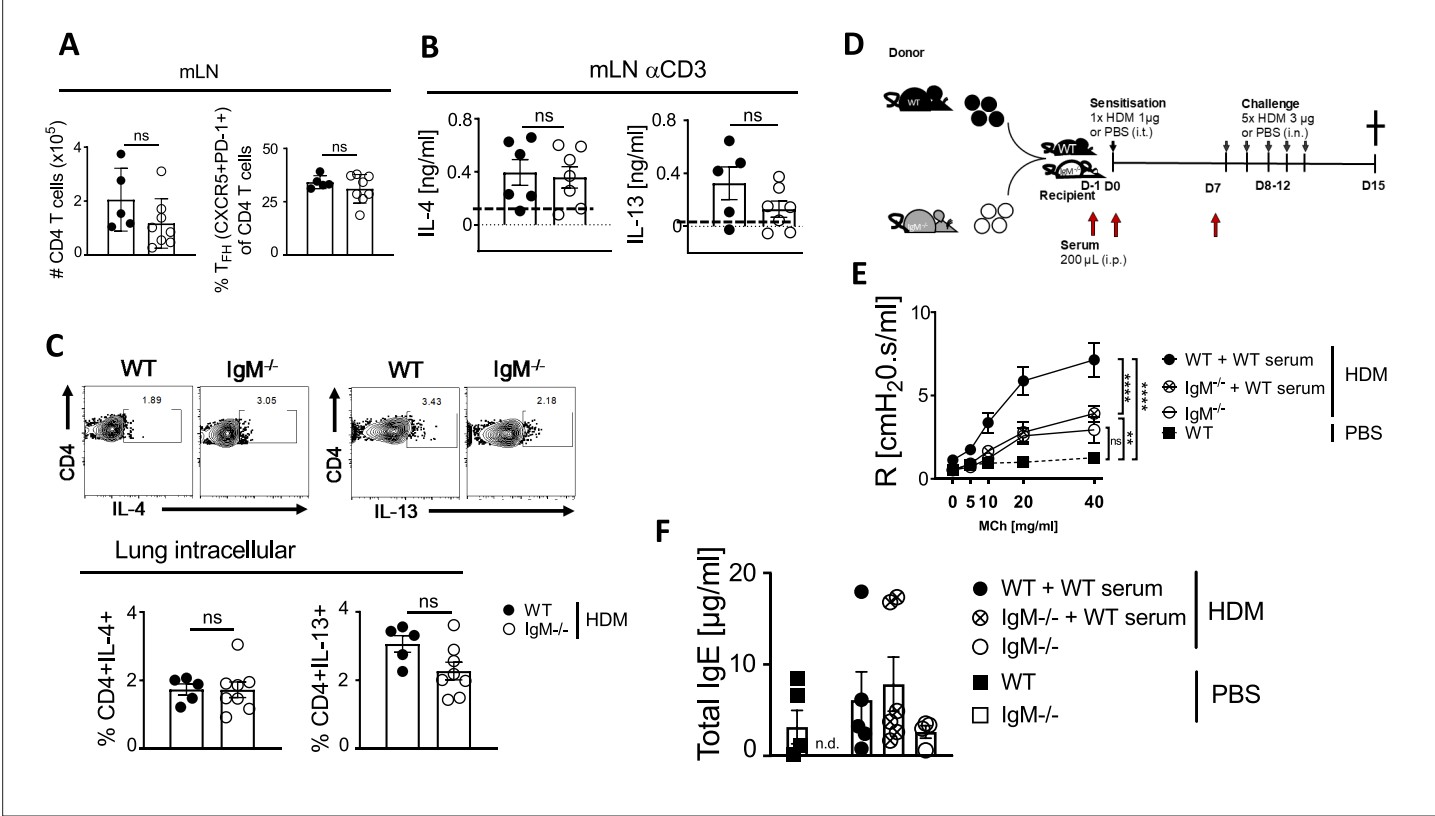

**Figure 2.** Immunoglobulin M (IgM)-deficiency does not lead to reduced T helper 2 (Th2) allergic airway inflammation and serum transfer restores IgE, but not airway hyperresponsiveness (AHR). Fig. a-c, mice treated as in *Figure 1A*. (**A**) Total mediastinal lymph node CD4 T cell numbers (live⁺CD3⁺CD4⁺) and % of Follicular Helper T cells (live⁺CD3⁺CD4⁺PD-1⁺CXCR5⁺) of CD4 T cells were stained and analysed by Flow cytometry and enumerated from % of live cells. (**B**) Mediastinal lymph nodes were stimulated with anti-CD3 (10 µg/mL) for 5 days and supernatants were used to measure levels of IL-4 and IL-13. Cytokines were not detected in unstimulated or HDM (30 µg) stimulated mediastinal lymph node (mLN). (**C**) Representative FACS plots and frequencies of lung CD4 T cells (live⁺CD3⁺CD4⁺) producing IL-4 and IL-13 after 5 hr stimulation with PMA/ionomycin in the presence of monensin. (**D**) Schematic diagram showing serum transfer from WT to IgM KO which were then sensitised as shown in *Figure 1*,A. (**E**) Airway resistance was measured with increasing doses of acetyl methacholine (0–40 mg/mL). (**F**) Total IgE in serum of mice either transferred with WT serum, IgM KO serum, or no serum. Shown is the mean ± SEM from two pooled experiments (n=5–8). Significant differences between groups were performed by Student *t*-test (Mann-Whitney) (**C, D, E**) or by two-Way ANOVA with Benferroni post-test (**B**) and are described as: *P<0.05, **Pp<0.01, ***p<0.001, ****p<0.0001.

The online version of this article includes the following figure supplement(s) for figure 2:

**Figure supplement 1.** B cell and bone marrow reconstitution do not restore airway hyperresponsiveness in Immunoglobulin M (IgM)-deficient mice in house dust mite (HDM)-induced allergic asthma.

cells were not affected by lack of IgM (*Figure 1G*, *Figure 1—figure supplement 1G*). The lack of class switching in Balb/C mice was also consistent with what we found in IgM KO mice in the C57BL/6 background (*Figure 1—figure supplement 3C*). This suggested a normal interaction between B and T cells in GCs, but lack of AID-dependent class switching, despite increased expression of IgD (*Figure 1—figure supplement 4A*; *Muramatsu et al., 2000*). We checked for natural IgM and antigen-induced IgM in multiple tissues. B cells expressing IgD were increased in all tissues, including mLNs, peritoneal cavity, and spleen (*Figure 1—figure supplement 4A–C*) and pre-B cell subsets were normal in BM (*Figure 1—figure supplement 4D*) in the absence of IgM as previously reported (*Lutz et al., 1998*).

## T helper 2 cells are intact in the absence of IgM, and serum transfer can partially restore antibody function

To investigate whether the reduction in AHR was due to reduced Th2 cells and cytokines, we stimulated total mLN and lung cells with anti-CD3 for 5 days or with PMA/ionomycin for 5 hr in the presence of monensin and measured secreted or intracellular IL-4 and IL-13 expression. We found no differences in CD4 T cells and T follicular helper cells (*Figure 2A*) in mLN and in secreted or intracellular levels of IL-4 and IL-13 between IgM KO and WT mice challenged with HDM in both mLN and lungs (*Figure 2B–C*). To investigate what drives this reduced AHR in the absence of IgM between secreted antibodies or the IgM B cell receptor on the surface of B cells, we transferred serum from naïve WT mice into IgM KO mice a day before sensitisation, during sensitisation and a day before challenge with HDM allergen (*Figure 2D*), as previously described (*Wojciechowski et al., 2009*). AHR in IgM KO was still reduced compared to WT mice even after the transfer of WT serum (*Figure 2E*), but levels of total IgE, HDM-specific IgE, and HDM-specific IgG1 were increased and comparable to those found in WT mice (*Figure 2F*, *Figure 2—figure supplement 1D*). The redundant role of IgE was consistent with previous studies where IgE nor its high-affinity receptor (FcεRI) were essential in AHR in HDM or OVA models (*McKnight et al., 2017*). The lack of functional role of serum-transferred IgE was consistent with earlier findings on *H. polygyrus* transfer of immune serum, where IgE was found not to be essential in protection against *H. polygyrus* re-infection (*Wojciechowski et al., 2009*).

## Replacement of IgM-deficient mice with functional hematopoietic cells in busulfan mice chimeric mice restores airway hyperresponsiveness

We then generated bone marrow chimeras by chemical radiation using busulfan (*Montecino-Rodriguez and Dorshkind, 2020*). We treated mice three times with busulfan for three consecutive days and after 24 hr transferred naïve bone marrow from congenic CD45.1 WT mice or CD45.2 IgM KO mice (*Figure 3A*, *Figure 3—figure supplement 1A*). We showed that recipient mice that did not receive donor bone marrow after 4 days post-treatment had significantly reduced lineage markers (CD45+Sca-1+) or lineage negative (Lin-) cells in the bone marrow when compared to untreated or vehicle (10% DMSO) treated mice (*Figure 3—figure supplement 1B–C*). We allowed mice to reconstitute bone marrow for 8 weeks before sensitisation and challenge with low dose HDM (*Figure 3A*). We showed that WT (CD45.2) recipient mice that received WT (CD45.1) donor bone marrow had higher airway resistance and elastance and this was comparable to IgM KO (CD45.2) recipient mice that received donor WT (CD45.1) bone marrow (*Figure 3B*). As expected, IgM KO (CD45.2) recipient mice that received donor IgM KO (CD45.2) bone marrow had significantly lower AHR compared to WT (CD45.2) or IgM KO (CD45.2) recipient mice that received WT (CD45.1) bone marrow (*Figure 3B*). We confirmed that the differences observed were not due to differences in bone marrow reconstitution, as we saw similar frequencies of CD45.1 cells within the lymphocyte populations in the lungs and other tissues (*Figure 3—figure supplement 1D*). We observed no significant changes in the lung neutrophils, eosinophils, inflammatory macrophages, CD4 T cells, or B cells in WT or IgM KO (CD45.2) recipient mice that received donor WT (CD45.1/CD45.2) or IgM KO (CD45.2) bone marrow when sensitised and challenged with low dose HDM (*Figure 3C*).

Restoring IgM function through adoptive reconstitution with congenic CD45.1 bone marrow in non-chemically irradiated recipient mice or sorted B cells into IgM KO mice (*Figure 2—figure supplement 1A*) did not replenish IgM B cells to levels observed in WT mice and as a result did not restore AHR, total IgE, and IgM in these mice (*Figure 2—figure supplement 1B–C*).

IgM is known to play a key role in shaping gut microbiota, and its diversity can be shaped by gut commensals (*Magri et al., 2017*; *Smith et al., 2023*). To understand whether microbiota could

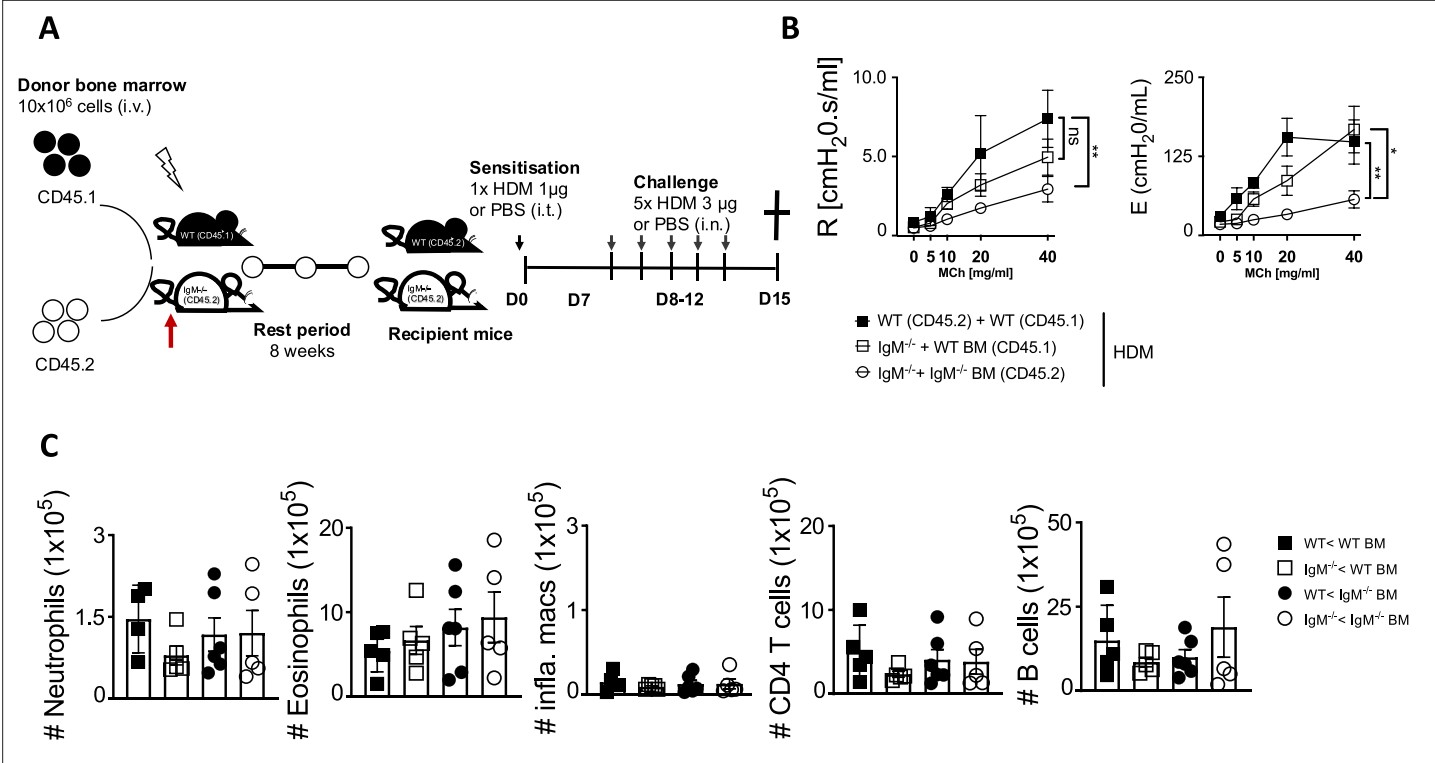

**Figure 3.** Partial wild-type bone marrow replenishment restores airway hyperresponsiveness (AHR) in Immunoglobulin M (IgM)-deficient mice (Figure A-C, mice treated as in *Figure 1A*). (**A**) Schematic diagram showing wild-type (WT) and IgM-deficient mice being chemically irradiated with busulfan (25 mg per day for 3 days) and adoptively transferred with congenic bone marrow (10×10⁶ per mouse intravenously) at day 4. Mice (WT to IgM KO plus bone marrow) were then rested for 8 weeks before being sensitised as shown in *Figure 1A*. (**B**) Airway resistance and elastance were measured with increasing doses of acetyl methacholine (0–40 mg/mL). (**C**) Total lung neutrophils (live⁺CD11b⁺Ly6G⁺), eosinophils (live⁺Siglec-F⁺CD11c⁻), inflammatory macrophages (live⁺Ly6G⁻CD11b⁺F4/80⁺), CD4 T cells (live⁺CD3⁺CD4⁺CD8⁻), and B cells (live⁺B220⁺CD19⁺MHCII⁺) were stained and analysed by flow cytometry and enumerated from % of live cells. Shown is the mean ± SD from one experiment (n=5–6 per group). Significant differences between groups were performed by Student *t*-test (Mann-Whitney) (**C**) or by two-way ANOVA with Benferroni post-test (**B**) and are described as: *$p<0.05$, **$p<0.01$, ***$p<0.001$, ****$p<0.0001$.

The online version of this article includes the following figure supplement(s) for figure 3:

**Figure supplement 1.** Bone marrow chimaera generation using busulfan to restore Immunoglobulin M (IgM) function.

**Figure supplement 2.** Reduced airway hyperresponsiveness in Immunoglobulin M (IgM)-deficient mice is independent of microbial influence in house dust mite (HDM)-induced asthma.

influence AHR in IgM-deficient mice, we treated WT and IgM KO with an antibiotic cocktail 3 x per week for 2 weeks (*Figure 3—figure supplement 2A*). We confirmed the reduction of bacteria in the faecal pellet after 2 weeks of antibiotic treatment (*Figure 3—figure supplement 2A*). We then sensitised and challenged these mice with HDM and measure AHR (*Figure 3—figure supplement 2B*). We observed increased resistance and elastance in WT mice compared to IgM KO mice and pre-treatment with antibiotics did not influence AHR (*Figure 3—figure supplement 2B*). Treatment with antibiotics did increase total IgE, HDM-specific IgE, and IgG1 in WT mice as expected (*Trompette et al., 2014*), but had no impact on IgM-deficient mice (*Figure 3—figure supplement 2C*), suggesting that overall antibiotics did not influence AHR.

## RNA sequencing reveals enrichment of genes associated with skeletal muscle contraction and actin re-arrangement

Because we had found no other changes in asthmatic allergic features between WT and IgM KO, except for profoundly reduced AHR which we could only restore when we partially replaced IgM-deficient haematopoietic cells with WT bone marrow, we resorted to RNA sequencing. We wanted to know if there were lung-specific factors that influenced AHR regulation in IgM-deficient mice. Principal

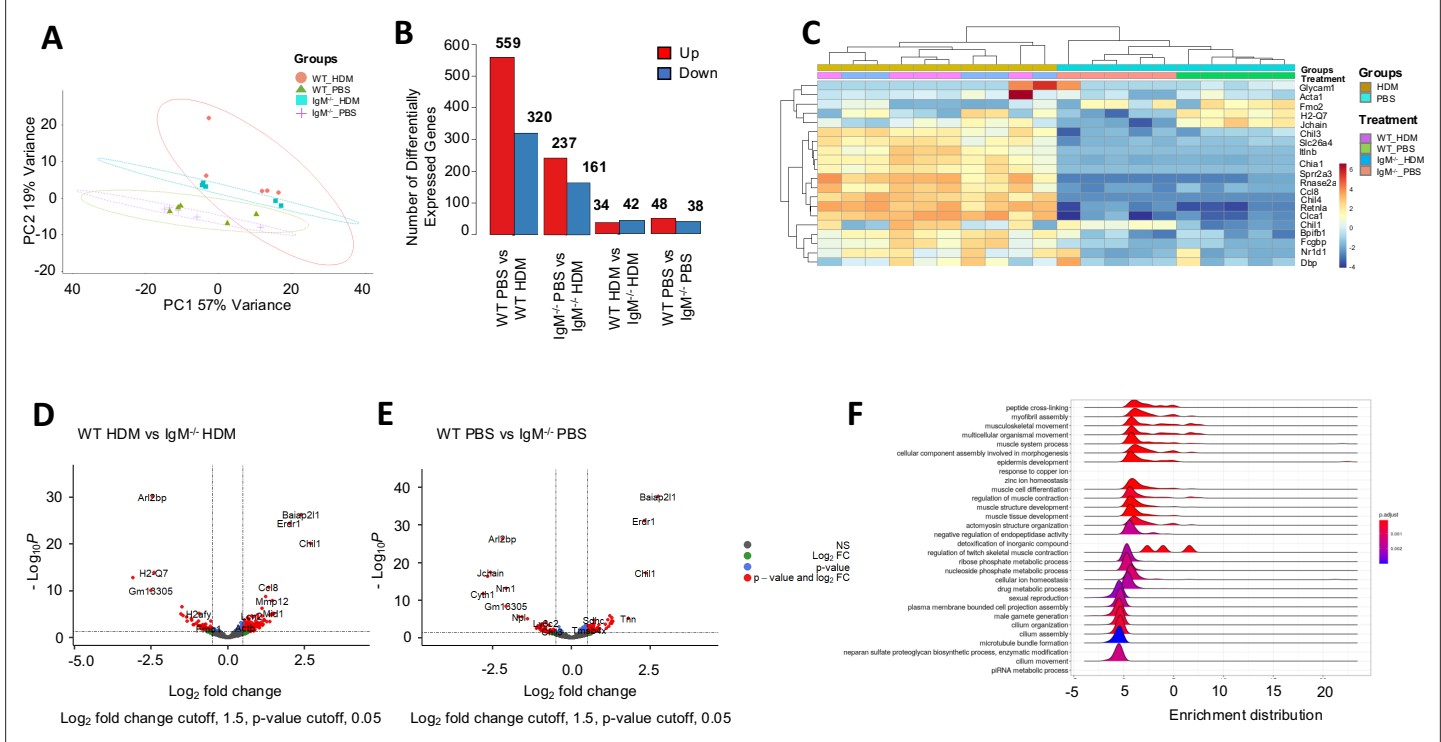

**Figure 4.** Genes associated with muscle contraction are downregulated in Immunoglobulin M (IgM)-deficient mice. Wild-type (WT) and IgM knockout (KO) mice were treated as in *Figure 1A* and RNA was collected from the whole lung for RNA sequencing. (**A**) Principal-component (PC) analysis showing variation in the global gene expression profiles across the different groups. PC1 (60%) and PC2 (18%), which capture the greatest variation in gene expression, are shown. Orange colour represents WT house dust mite (HDM), green colour represents WT PBS, blue colour represents IgM KO HDM and purple crosses represent IgM KO PBS. Each dot represents an individual mouse. (**B**) Number of differentially expressed genes between groups. (**C**) Heatmaps depicting the differently expressed genes between WT and IgM KO samples from HDM-treated and PBS mice ranked based on hierarchical clustering. (**D–E**) Volcano plots: numbers and colour relate to genes that have an adjusted *p*-value <0.05. Blue, significantly downregulated; red, significantly upregulated; grey, non-differentially expressed. P values were adjusted for multiple testing using the Benjamini-Hochberg method. (**D**) represent changes between WT and IgM KO treated with HDM and (**E**) represents changes between WT and IgM KO treated with saline. (**F**) Gene set enrichment analysis (GSEA) of hallmark gene sets from the Molecular Signatures Database of the Broad Institute, showing the normalized enrichment scores (NES) for lung RNA-Seq data from WT mice.

The online version of this article includes the following figure supplement(s) for figure 4:

**Figure supplement 1.** Genes associated with muscle contraction are downregulated in Immunoglobulin M (IgM)-deficient mice.

component analysis (PCA) depicted distinct global transcriptional changes with PC1 and PC2 explaining most of the variation (*Figure 4A*). We found a smaller variation in gene expression between WT and IgM KO in both HDM-challenged and PBS control mice (*Figure 4B*). We could mainly detect downregulation of genes such as brain-specific angiogenesis inhibitor 1-associated protein 2-like protein 1 (*Baiap2l1*), erythroid differentiation regulatory factor 1 (*Erdr1*), chemokines such as *Ccl8*, *Ccl9*, *Ccl17* and *Ccl22* in IgM KO mice challenged with HDM (*Figure 4C–D*). Interestingly, *Baiap2l1* and *Erdr1* were also downregulated in IgM KO saline-treated control mice, suggesting an inflammation-independent effect (*Figure 4C and E*). We also found the presence of the J-chain coding gene in WT mice which was absent in IgM KO mice, confirming a deletion in genes associated with holding IgM monomers together and thus IgM (*Norderhaug et al., 1999*; *Figure 4E*). Gene set enrichment analyses (GSEA) confirmed that the genes contributing to changes in AHR between WT and IgM KO were associated with muscle system processes and skeletal muscle contraction (*Figure 4F*). There was also an over-representation of gene ratios associated with skeletal muscle development, differentiation and contraction, and suppression of genes associated with chemotaxis and plasma membrane-bounded cell projection (*Figure 4—figure supplement 1*).

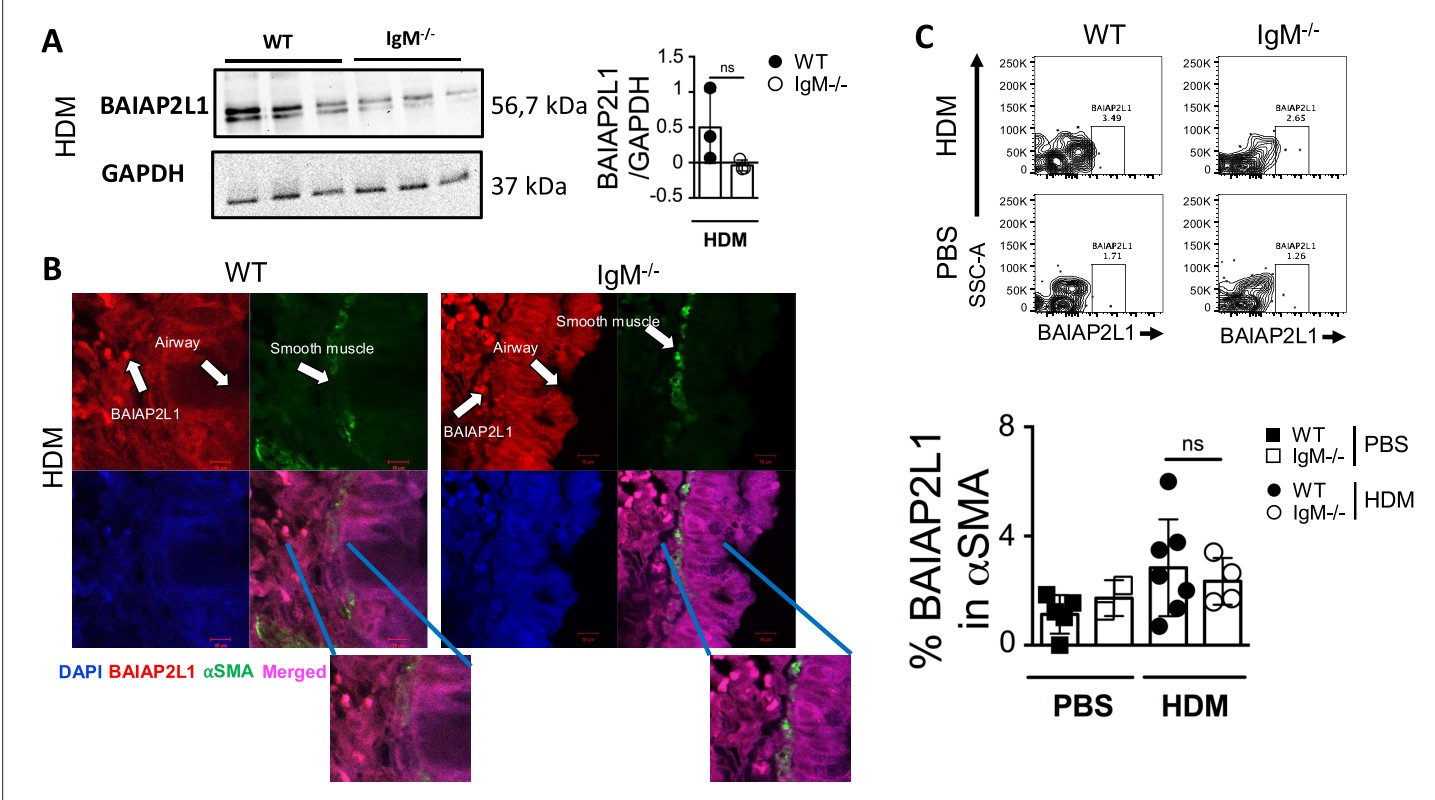

**Figure 5.** *BAIAP2L1* is expressed in close contact to smooth muscle. (**A**) Mouse lungs were homogenised in RIPA buffer and blotted on nitrocellulose. Rabbit anti-human BAIAP2L1 and mouse anti-GAPDH was used as primary antibody. Lines 1–3 is wild-type (WT) mice, lines 4–6 is Immunoglobulin M (IgM) knockout (KO) mice sensitised and challenged with house dust mite (HDM). (**B**) Lung sections from WT and IgM KO mice sensitised and challenged with HDM were immunostained for nuclei stained DAPI (Blue), anti-BAIAP2L1 (red), α smooth muscle actin (green), and merged images (Magenta). Insert below shows a zoomed-in image of merged BAIAP2L1 and α smooth muscle actin. (**C**) Representative flow cytometry plots showing BAIAP2L1 expression (Live⁺Singlets⁺CD45⁻α SMA⁺BAIAP2L1⁺) in WT and IgM KO treated with HDM or PBS. Quantification of % BAIAP2L1 in α smooth muscle actin is shown. Shown are representative images from two independent experiments (n=3 mice per group).

The online version of this article includes the following source data and figure supplement(s) for figure 5:

**Source data 1.** Western blot showing expression of BAIAP2L1 at 56.7kDa in WT and IgM KO mouse lungs and control GAPDH at 37 kDa.

**Source data 2.** Western blot showing raw data BAIAP2L1 and GAPDH in WT and IgM KO mouse lungs.

**Figure supplement 1.** BAIAP2L1 is mainly expressed by alpha-smooth muscle cells.

## *BAIAP2L1* is expressed in lung cells in contact with airway smooth muscle cells

We decided to focus on *Baiap2l* also known as Inverse-bin-amphiphysin-Rvs (I-BAR)-domain-containing protein insulin receptor tyrosine kinase substrate (IRTKS), which has been shown to promote actin polymerisation and microvilli length in the intestinal epithelial cells (*Postema et al., 2018*). We first verified whether BAIAP2L1 was also downregulated at the protein level in IgM-deficient mouse lungs challenged with HDM (*Figure 5A*). We found lower expression levels of BAIAP2L1 in IgM-deficient mice compared to WT mice challenged with HDM by western blotting, although this was not significant (*Figure 5A*). Human Protein Atlas search suggested that BAIAP2L1 is expressed by multiple cell types, including fibroblasts, skeletal muscle, smooth muscle, and macrophages (*Abo et al., 2020*). We then investigated by immunofluorescence which cell types within the lung could be expressing BAIAP2L1 (*Figure 5B*). We found BAIAP2L1 to be closely expressed within structural cells in close contact to smooth muscle cells (less than 10 µm), a cell type known to be essential in bronchoconstriction. Airway smooth muscle together with the extracellular matrix contributes to smooth muscle hypertrophy which causes the narrowing of the airways. These cells, by histology, appear to be separated due to fixation methods but are mashed together, especially during the remodelling of the airways (*James et al., 2012*). IgM deficiency did not impact

this expression in cells in close contact with smooth muscle (*Figure 5B*). Because both western blot and immunofluorescent were inconclusive, we also verified the expression of BAIAP2L1 via flow cytometry (gating in *Figure 5—figure supplement 1A*) and found it to be expressed mainly by alpha-smooth muscle cells and increased in expression in HDM-treated mice compared to PBS control mice (*Figure 5C*). Furthermore, we showed a reduction in BAIAP2L1 expression amongst alpha-smooth muscle cells but not other cells (CD45 positive or alpha-smooth muscle negative cells, *Figure 5—figure supplement 1B*) in IgM-deficient mice compared to WT mice treated with HDM, although this did not reach significance (*Figure 5C*). Overall, this data suggested that BAIAP2L1 is reduced at RNA and protein level and likely influenced the reduction of AHR in IgM-deficient mice. This was consistent with gene set enrichment in our RNA seq data, where there was a dominance of genes associated with muscle contraction which regulates AHR.

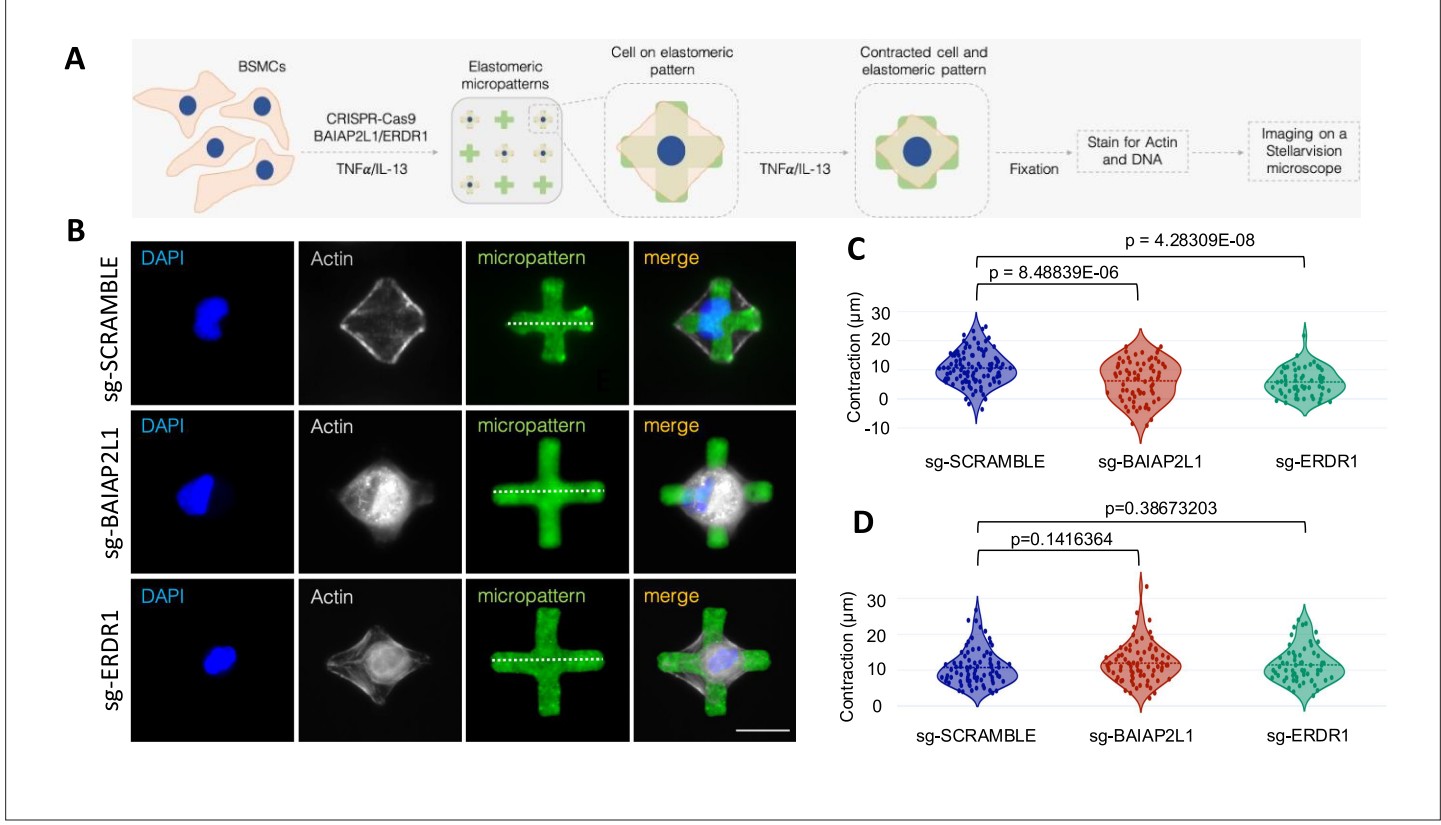

**Figure 6.** CRISPR-based deletion of *BAIAP2L1* leads to reduced smooth muscle contraction at a single-cell level. (**A**) Bronchial smooth muscle cells (1.6×10$^5$ cells/well) were transfected with CRISPR-Cas9 single guide RNAs (scramble, *BAIAP2L1*, and *ERDR1*), stimulated with recombinant human IL-13 (100 ng/mL) and TNF-a (10 ng/mL) for 48 hr. Cells were then transferred to elastomeric micropatterns, stimulated again with rIL-13 and rTNF-a and fixed before imaging on a StellarVision microscope. (**B**) Representative images of single bronchial smooth muscle cells (BSMCs) on micropatterns from scramble, *BAIAP2L1,* and *ERDR1* stimulated with 10 ng/mL TNF-a. DNA was stained with DAPI, actin fibers with Phalloidin-565 and elastomeric micropatterns are coated in Fibronectin-488. Merged images are shown on the right. (**C**) Violin plots showing contraction of 50–100 cells/condition stimulated with 10 ng/mL TNF-a, individual dots represent a single cell contraction, where blue is scramble sgRNA, red is *BAIAP2L1* sgRNA, and green is *ERDR1* is sgRNA. (**D**) Violin plots showing contraction of 50–100 cells/condition stimulated with 100 ng/mL IL-13, individual dots represent a single cell contraction, where blue is scramble sgRNA, red is *BAIAP2L1* sgRNA, and green is *ERDR1* is sgRNA. Shown is mean ± SEMs from two pooled experiments (n=50–100). Significant differences between groups were performed by the Student t-test (Mann-Whitney) and *p*-value is shown.

The online version of this article includes the following source data and figure supplement(s) for figure 6:

**Figure supplement 1.** Targeting and validation of *BAIAP2L1* deletion by CRISPR.

**Figure supplement 1—source data 1.** DNA gel showing BAIAP2L1 expression in bronchial smooth muscle cells stimulated with acetylcholine and TNF-alpha and either transfected with sgRNA scramble or sgRNA BAIAP2L1.

**Figure supplement 1—source data 2.** DNA gel showing BAIAP2L1 expression in bronchial smooth muscle cells stimulated with ACh and TNF-alpha and transfected with sgRNA scramble or sgRNA BAIAP2L1.

## CRISPR-Cas9 deletion of *BAIAP2L1* leads to reduced airway smooth muscle contraction at a single cell

To understand how *Baiap2l1*, one of the genes downregulated in IgM-deficient mice, could influence AHR, we resorted to an in vitro model that allowed us to measure contraction using fluorescently labelled elastomeric contractible surface (FLECS) technology (*Pushkarsky et al., 2018*; *Figure 6A*). We opted for airway smooth muscle as it has been shown to contribute to the narrowing of the airway during an allergic attack and together with neighbouring cells, such as extracellular matrix and immune cells are critical in hypertrophy (*James et al., 2012*). We chose the human bronchial smooth muscle cell (BSMC) line as *BAIAP2L1* is expressed in structural cells, including muscle and epithelial cells, and actin together with myosin is essential in muscle contraction (*Abo et al., 2020*). To this end, we used CRISPR-Cas9 technology to knock down *BAIAP2L1* in BSMCs (*Figure 6—figure supplement 1A–B*). We also included ERDR1 as one of the genes that were also downregulated in IgM-deficient mice. BSMCs (160,000/well) were stimulated with 10 ng/mL of human TNF-α and 100 ng/mL IL-13, followed by transfection with ribonucleoprotein (RNP) complexes containing single guide RNA targeting *BAIAP2L1* and Cas9 for 48 hr (*Figure 6A*). Transfected cells were seeded onto elastomeric patterns, stimulated with the same stimulants for 3 hr, and fixed in 4% paraformaldehyde, followed by staining with DAPI and Phalloidin before imaging on Stellarvision microscope and quantification (*Figure 6A*). We validated the deletion of *BAIAP2L1* by PCR and showed reduced expression of *BAIAP2L1* in acetylcholine (ACh) and TNF-α stimulated cells transfected with *BAIAP2L1* sgRNA when compared to scramble sgRNA (*Figure 6—figure supplement 1C*, line 3 and 4 compared to line 5 and 6). We

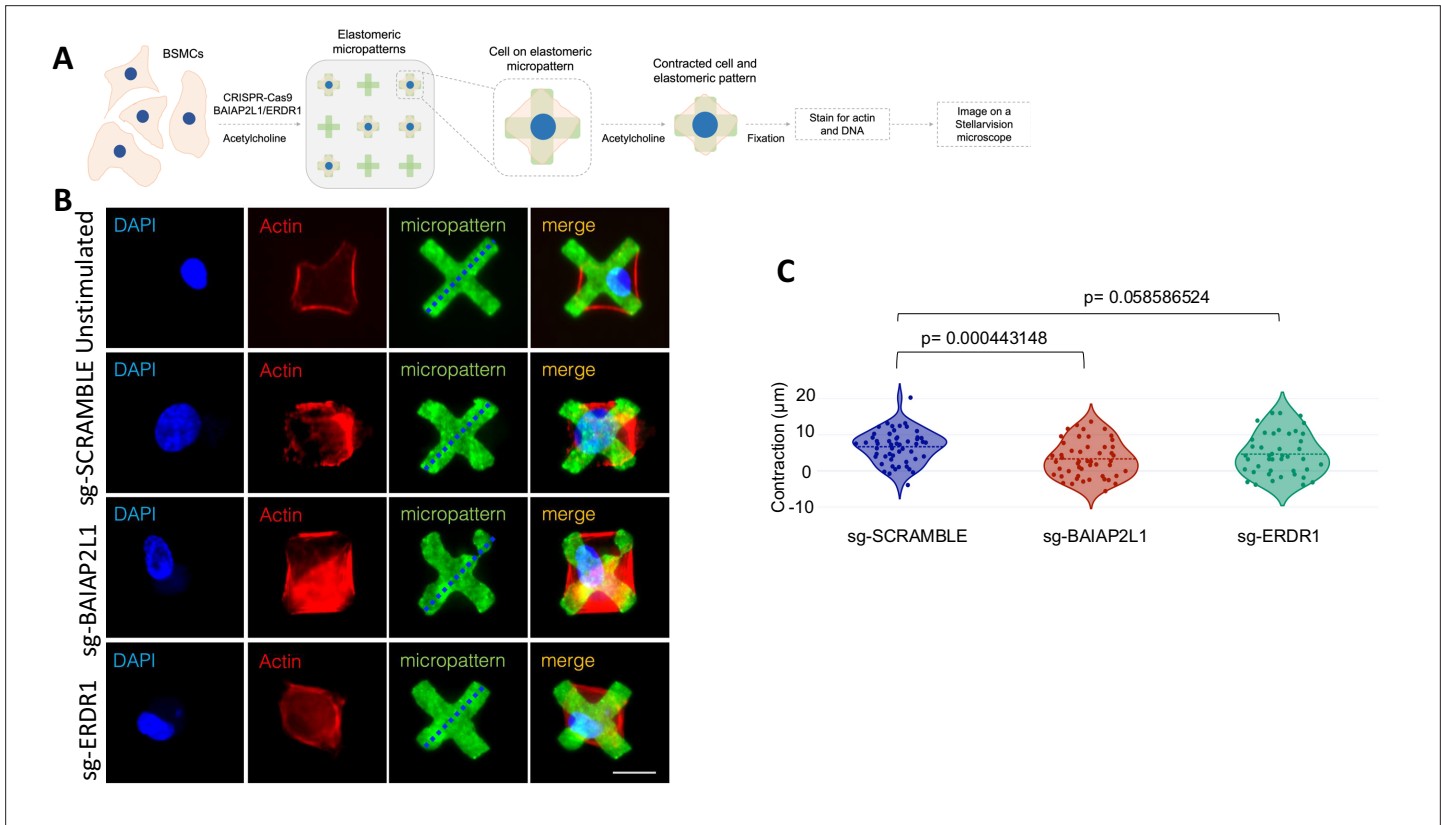

**Figure 7.** CRISPR-based deletion of *BAIAP2L1* reduces smooth muscle contraction upon stimulation with acetylcholine. (**A**) Bronchial smooth muscle cells ($1.6 \times 10^5$ cells/well) were transfected with CRISPR-Cas9 single guide RNAs (scramble, *BAIAP2L1*, and *ERDR1*), stimulated with Acetylcholine (10 µM) for 48 hr. Cells were then transferred to elastomeric micropatterns, stimulated again with ACh (10 µM) and fixed before imaging on a StellarVision microscope. (**B**) Representative images of single bronchial smooth muscle cells (BSMCs) on a single micropattern from unstimulated, scramble, *BAIAP2L1*, and *ERDR1* stimulated with ACh (10 µM). Shown are DAPI, actin (red), a green fluorescent micropattern and merged images. (**C**) Violin plots showing contraction of 50–100 cells/condition stimulated with ACh (10 µM) individual dots represent a single cell contraction, where blue is scramble sgRNA, red is BAIAP2L1 sgRNA, and green is ERDR1 is sgRNA. Shown is mean ± SD from one representative experiment of two independent experiments (n=50–100). Significant differences between groups were performed by the Student t-test (Mann-Whitney) and *p*-value is shown.

also detected low expression of BAIAP2L1 in unstimulated sgRNA scramble and sgRNA *BAIAP2L1*-transfected cells (*Figure 6—figure supplement 1C*, lines 1 and 2). Transfections with sgRNA did not impact cell viability (*Figure 6—figure supplement 1D*). As expected, BSMCs displayed a higher level of contraction in TNF-α or IL-13 stimulated cells compared to unstimulated cells (*Figure 6B–D*). In BSMCs transfected with sgRNA-*BAIAP2L1* and -*ERDR1*, we saw a significant reduction in contraction levels in TNF-α-stimulated cells when compared to scramble sgRNA-transfected and stimulated cells (*Figure 6B and C*). We saw no differences in contraction between scramble sgRNA and sgRNA-*BAIAP2L1* and -*ERDR1* stimulated with IL-13 (*Figure 6D*).

## *BAIAP2L1* deletion but not *ERDR1* is essential in acetylcholine-induced smooth muscle contraction

Contraction of the ASM is central to AHR as it controls airway diameter and airflow which can increase resistance (*Lambrecht and Hammad, 2015*). Airway smooth muscle contractility is induced when extracellular factors such as ACh bind through muscarinic 3 acetylcholine receptor (m$_3$AChR), activating a signalling cascade leading to calcium accumulation and contraction (*Ouedraogo and Roux, 2014*). We transfected BSMCs with sgRNA-*BAIAP2L1*, -*ERDR1*, or scrambled and stimulated cells with 10 μM ACh (*Figure 7A*). We saw a significant reduction in BSMC contraction transfected with sgRNA-*BAIAP2L1* when compared to scrambled sgRNA-transfected and ACh-stimulated (*Figure 7B–C*). We saw no significant differences in BSMC contraction between sgRNA-*ERDR1*-transfected and scramble sgRNA-transfected and ACh-stimulated cells (*Figure 7B–C*). Taken together, this data indicates that *BAIAP2L1*, a gene that was downregulated in IgM-deficient mice has a stimulant-specific role in inducing airway contraction of bronchial smooth muscle cells during asthma.

## Discussion

Here, we described an unexpected function of IgM in the regulation of bronchoconstriction. We show by RNA sequencing that in the lung tissue of IgM-deficient mice, there is a reduction in genes (*Baiap2l1* and *Erdr1*) associated with actin cytoskeleton and re-arrangement, a key factor in smooth muscle contraction and AHR. We used single-cell force cytometry and CRISPR-Cas9 technology to validate these genes in the human BSM cell line and show, as a proof of concept, that deletion of *BAIAP2L1* reduced muscle contraction in a stimulant-specific manner.

B cells play a complex role in allergic asthma, earlier studies using B cell-deficient mice showed a redundant role of B cells in OVA-induced allergic airway inflammation and airway hyperreactivity (*Hamelmann et al., 1999*; *Korsgren et al., 1997*; *MacLean et al., 1999*). We and others have recently shown using a more complex allergen HDM-relevant to human asthma that antigen load is crucial in B cell function (*Dullaers et al., 2017*; *Habener et al., 2021*; *Hadebe et al., 2021*). We explored these possibilities in the context of IgM deficiency and observed a profound reduction in AHR. This was in complete contrast to what has been observed in B cell-deficient mice and in IgE or FcεRI-deficient mice (*Dullaers et al., 2017*; *Habener et al., 2021*; *McKnight et al., 2017*). We also found a similar reduction in AHR at higher doses of HDM, albeit less pronounced. Reduction in AHR was not only specific to HDM, but we also found reduced AHR in IgM-deficient mice sensitised with OVA complexed to alum and challenged with OVA, although less pronounced. AHR reduction in IgM-deficient mice was also seen in the acute papain model which only activates innate responses, including innate lymphoid cells (*Darby et al., 2021*). Interestingly, this reduction in AHR was not associated with reduced allergic airway inflammation, including eosinophilia, mucus production, or Th2 cells in all models tested, which was unexpected considering these features are key in AHR. B cell-deficient mice show a reduction in eosinophilia and antigen-specific Th2 cells at low doses of HDM, which leads to reduced AHR (*Dullaers et al., 2017*). Similar findings are observed in mice lacking IL-4Rα specifically in B cells when challenged with HDM (*Hadebe et al., 2021*).

IgM deficiency did not impact B cell development in primary and secondary lymphoid tissues and accumulation of follicular and germinal centre B cells. Interestingly, there was a lack of class switching to IgG1 and IgE isotypes despite increased expression of IgD. The lack of class switching to IgG1 and IgE contrasts with earlier reports, which suggested that IgD can largely replace IgM for class switching to other isotypes, resulting in delayed neutralising IgG1 against vesicular stomatitis virus (VSV) (*Lutz et al., 1998*; *Ochsenbein et al., 1999*). To get a better understanding of which IgM function was

responsible for AHR, we showed that the transfer of naïve wild-type mice serum to IgM-deficient mice could restore IgE production and HDM-specific IgE and IgG1. This is consistent with what has been observed in the context of influenza viral infections, where the transfer of purified or serum IgM into B cell-deficient mice restored IgM-induced viral neutralisation (*Jayasekera et al., 2007*). We believe that naïve sera transferred to IgM-deficient mice were able to bind to the surface of B cells via IgM receptors (FcμR/Fcα/μR), which are still present on IgM-deficient B cells, and this signalling is sufficient to facilitate Class Switch Recombination (CSR). This was also confirmed by the ability of B cells to reach germinal centres in IgM-deficient mice during HDM exposure. Our IgM KO mouse lacks both membrane-bound and secreted IgM and transferred serum contains at least secreted IgM which can bind to B cell surfaces. Of course, we can't rule out that transferred sera from WT mice also contains some IgG1 which can facilitate class switching to IgE when transferred to IgM-deficient mice. Despite the ability of wild-type serum to restore IgE, it was not enough to restore AHR, which may be due to the difficulties in restoring IgM to normal levels (200–800 μg). Our findings on the redundant role of IgE were consistent with previous studies where IgE nor its high-affinity receptor (FcεRI) were essential in AHR in an HDM or OVA model (*McKnight et al., 2017*). The lack of functional role of serum-transferred IgE was consistent with earlier findings on *H. polygyrus* transfer of immune serum, where IgE was found not to be essential in protection against *H. polygyrus* re-infection (*Wojciechowski et al., 2009*). To resolve other endogenous factors that could have potentially influenced reduced AHR in IgM-deficient mice, we resorted to busulfan chemical irradiation to deplete bone marrow cells in IgM-deficient mice and replace bone marrow with WT bone marrow. While it is well accepted that busulfan chemical irradiation partially depletes bone marrow cells, in our case, it was not possible to pursue other irradiation methods due to changes in ethical regulations and the fact that mice are slow to recover after gamma rays irradiation. Busulfan chemical irradiation allowed us to show that we could mostly restore AHR in IgM-deficient recipient mice that received donor WT bone marrow when challenged with low dose HDM.

B cell isotype IgD has been shown to promote Th2 cells and OVA-, papain-, and NP-specific humoral responses, specifically IgG1 and IgE, through basophil activation (*Shan et al., 2018*). IgD has also been shown to inhibit IgE-mediated basophil activation through downregulation of genes associated with cytoskeleton organisation, such as signal transducers phosphoinositide-3 kinase, RAS, and RHO (*Shan et al., 2018*). Our RNA sequencing data showed downregulation of *Baiap2l1* and *Erdr1* in IgM-deficient mice, genes which have been shown to associate with actin cytoskeleton and re-arrangement (*Abo et al., 2020*; *Houh et al., 2016*; *Postema et al., 2018*). Contraction of the ASM is central to AHR as it controls airway diameter and airflow which can increase resistance (*Perkins et al., 2011*). ASM activated by cytokines such as IL-13 or TNF-α or bronchoconstrictor ACh induces cytosolic calcium accumulation (*Moulton and Fryer, 2011*; *Perkins et al., 2011*). $Ca^{2+}$ release results in a cascade of events that involves kinases and phosphorylation of key molecules such as actin and myosin involved in smooth muscle contraction (*Ouedraogo and Roux, 2014*). Gene set enrichment analysis suggested that genes downregulated in IgM-deficient mice were involved in processes such as muscle contraction, muscle twitching, and skeletal muscle movement. As a proof of concept, we validated the involvement of *BAIAP2L1* in bronchial smooth muscle contraction using a high-throughput method that allows us to measure the contraction of 1000s of single cells (*Pushkarsky et al., 2018*). When we deleted *BAIAP2L1* using CRISPR-Cas9, we showed that deletion of *BAIAP2L1* in BSMCs reduced smooth muscle contraction when stimulated with TNF-α and ACh, but not IL-13. Deletion of ERDR1 had a minor impact on muscle contraction across different stimulants suggesting a stimulant and gene-specific role. IRTKS/BAIAP2L1 has been shown to regulate microvilli elongation by forming a complex with actin-regulating protein EPS8 (*Postema et al., 2018*). In this setting, IRTKS elongate microvilli via distinct mechanisms that require functional WH2 and SH3 domains, which binds EPS8, an F-actin capping and bundling protein (*Postema et al., 2018*). We have not fully dissected how *BAIAP2L1* could be regulated by IgM, but it is clear that through its actions to regulate actin bundling, it is involved in muscle contraction which requires actin and myosin head interaction (summarised in *Figure 8*).

We speculate that IgM can directly activate smooth muscle cells by binding a number of its surface receptors, including FcμR, Fcα/μR, and pIgR (*Liu et al., 2019*; *Nguyen et al., 2017b*; *Shibuya et al., 2000*). IgM binds to FcμR strictly, but shares Fcα/μR and pIgR with IgA (*Liu et al., 2019*; *Michaud et al., 2020*; *Nguyen et al., 2017b*). Both Fcα/μR and pIgR can be expressed by non-structural cells at mucosal sites (*Kim et al., 2014*; *Liu et al., 2019*). We would not rule out that the mechanisms of muscle contraction might be through one of these IgM receptors, especially the ones expressed on smooth muscle cells

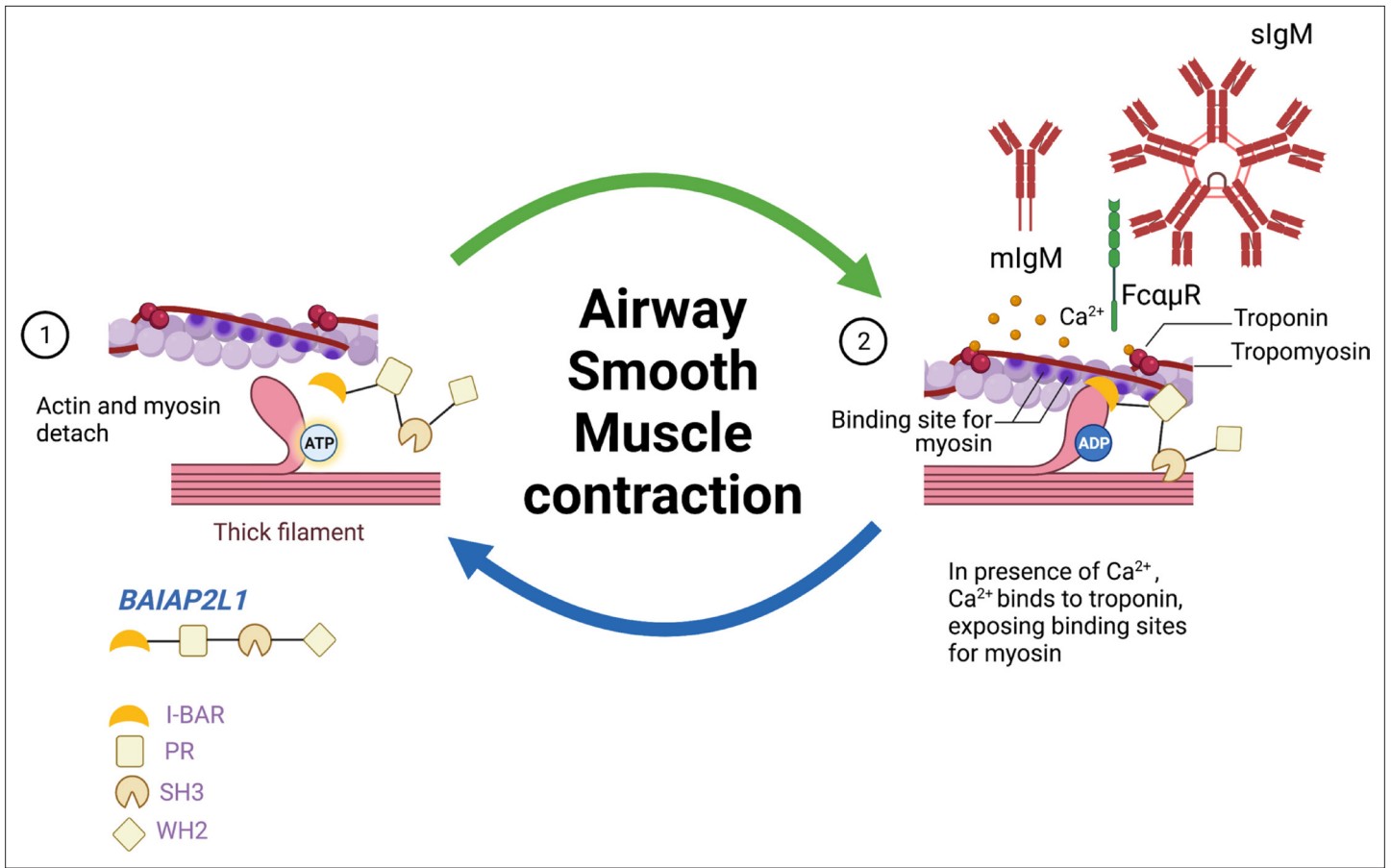

**Figure 8.** Working model showing how *Baiap2l1* could influence muscle contraction through influencing myosin and actin filament interaction.

(*Kim et al., 2014*; *Liu et al., 2019*). Certainly, our future studies will be directed towards characterizing the mechanism by which IgM potentially activates the smooth muscle.

We believe these findings report for the first time an independent function of IgM from its natural class switching and non-specific binding to microbial components. We believe that IgM function in regulating ASM may be indirect through other unknown factors but does not involve microbiota, as treatment of mice with a mixture of antibiotics did not restore AHR. Early vaccination with bacterial species, such as group *A streptococcus* containing GlcNAc or β–1,3-glucans can protect adult mice against *A. fumigatus*-induced allergic asthma (*Kin et al., 2012*). This is mainly through conserved germline-encoded IgM antibodies, which have broad specificities to common allergens containing GlcNAc moieties such as dermatophytes (*Kearney et al., 2015*; *New et al., 2020*). Together, these findings demonstrate for the first time an important function of IgM in regulating airway hyperresponsiveness independent of the presence of T helper 2 allergic inflammation. These findings have implications for future treatment of allergic asthma through bronchodilators.

## Materials and methods
### Mice

IgM-deficient homozygote mice on Balb/C background and on C57BL/6 background were originally described in *Lutz et al., 1998* and were backcrossed at least 10 generations in house at the University of Cape Town (*Lutz et al., 1998*). Wild-type on Balb/C (The Jackson Laboratory, RRID:IMSR_JAX:000651) and C57BL/6 (The Jackson Laboratory, RRID:IMSR_JAX:000664) backgrounds were used as a littermate control. Congenic wild-type Balb/C mice (CByJ.SJL(B6)-Ptprca/J, RRID:IMSR_JAX:006584) were

backcrossed to at least 10 generations at the University of Cape Town. B cell-deficient mice (muMt B6.129S2-Ighmtm1Cgn/J, RRID:IMSR_JAX:002288) was originally described in *Kitamura et al., 1991* and was backcrossed at least 10 generations to a Balb/C background at the University of Cape Town. Mice were housed in independently ventilated cages under specific pathogen-free conditions at the University of Cape Town Animal Facility. All mice were used at 8 to 10 weeks of age and animal procedures were performed according to the strict recommendation by the South African Veterinary Council and were approved by the University of Cape Town Animal Ethics Committee (Reference numbers 014/019, 018/013, and 022/014).

## House dust-mite induced allergic airway disease

A high dose and a low dose HDM treatment schedule were used to induce symptoms of allergic asthma in mice (*Hadebe et al., 2021*). Mice were anaesthetised with ketamine (Anaket-V; Centaur Labs, Johannesburg, South Africa) and xylazine (Rompun; Bayer, Isando, South Africa). For low-dose HDM, mice were sensitised intratracheally (i.t.) on day 0 with 1 µg of HDM (Stellergens Greer Laboratories, Lenoir, U.S.A.) and intranasally challenged with 3 ug HDM on days 8, 9, 10, 11, and 12. For high-dose HDM, mice were with HDM 100 ug and challenged with HDM 10 ug. AHR was measured on day 15. After the procedure, mice were euthanised and tissue samples were collected for analysis.

## Adoptive transfer of naïve B cells

Spleens were collected from naïve congenic CD45.1 Balb/C mice and passed through 40 µm strainer to obtain single-cell suspensions. Cells were stained with FITC-B220 and APC-CD19 for 30 min at 4 °C. A dead cell exclusion dye (7AAD) was added before sorting on BD FACS Aria I to at least 96% purity. 2–5 ×$10^6$ cells were adoptively transferred intravenously (i.v.) into IgM KO recipient mice a day before HDM sensitisation.

## Adoptive transfer of naïve serum

Naïve wild-type mice were euthanised and blood was collected via cardiac puncture before being spun down (5500 rpm, 10 min, RT) to collect serum. Serum (200 µL) was injected intraperitoneally into IgM-deficient mice. Serum was injected intraperitoneally at day –1, 0, and a day before the challenge with HDM (day 10).

## Busulfan bone marrow chimeras

WT (CD45.2) and IgM KO (CD45.2) congenic mice were treated with 25 mg/kg busulfan (Sigma-Aldrich, Aston Manor, South Africa) per day for three consecutive days (75 mg/kg in total) dissolved in 10% DMSO and Phosphate-buffered saline (0.2 mL, intraperitoneally) to ablate bone marrow cells. Twenty-four hours after last administration of busulfan, mice were injected intravenously with fresh bone marrow (10×$10^6$ cells, 100 µL) isolated from hind leg femurs of either WT (CD45.1) or IgM KO mice (*Montecino-Rodriguez and Dorshkind, 2020*). Animals were then allowed to complement their haematopoietic cells for 8 weeks. In some experiments, the level of bone marrow ablation was assessed 4 days post-busulfan treatment in mice that did not receive donor cells. At the end of the experiment level of complemented cells were also assessed in WT and IgM KO mice that received WT (CD45.1) bone marrow.

## Ovalbumin-induced allergic airway inflammation

Mice were sensitised intraperitoneally with (50 µg in 200 µl) of ovalbumin (OVA) adsorbed to 0.65% alum (Sigma-Aldrich, Aston Manor, South Africa) on days 0, 7, 14. On days 23, 24, and 25, mice were intranasally challenged with 100 µg of OVA under anaesthesia with ketamine (Anaket-V; Centaur Labs, Johannesburg, South Africa) and xylazine (Rompun; Bayer, Isando, South Africa). AHR was measured on day 26. After the procedure, mice were euthanized with halothane and tissue samples collected for analysis.

## Papain-induced lung inflammation

Mice were anaesthetized with isoflurane (3 L/min) briefly before being challenged with 50 µL of 25 µg of Papain (Sigma-Aldrich, Aston Manor, South Africa) on days 1, 2, and 3 or PBS. AHR was measured on day 4. After the procedure, mice were euthanized with halothane and tissue samples collected for analysis.

## Airway hyperresponsiveness

Airway resistance and elastance of the whole respiratory system (airways, lung, chest wall) after intranasal challenge was determined by forced oscillation measurements as described previously (*Kirstein et al., 2016*) with the Flexivent system (SCIREQ, Montreal, Canada) by using the single compartment ('snapshot') perturbation. Measurements were carried out with increasing doses of acetyl-β-methylcholine (methacholine, Sigma-Aldrich, Aston Manor, South Africa) (0, 5, 10, 20, and 40 mg/mL) for Balb/C or (0, 20, 40, 80, 160, and 320 mg/mL) for C57BL/6. Differences in the dose-response curves were analysed by repeated-measures two-way ANOVA with the Bonferroni post-test. Only mice with acceptable measurements for all doses (coefficient of determination >0.90) were included in the analysis.

## Flow cytometry

Single-cell suspensions were prepared from lymph nodes in Roswell Park Memorial Institute (RPMI) media (Gibco, Paisley, United Kingdom) by passing them through 100 μm strainers. To obtain single-cell suspensions from lung tissues, a left lobe was digested for 1 hr at 37 °C in RPMI containing 13 mg/mL DNase I (Roche, Randburg, South Africa) and 50 U/mL collagenase IV (Gibco, Waltham, Massachusetts) and passed through 70 μm strainer. Antibodies used in these experiments included, phycoerythrobilin (PE)-conjugated anti-Siglec-F (clone, E50-2440), anti-IL-5 (clone, TRFK5), anti-CD44 (clone, KM114), anti-T and B cell activation antigen (clone, GL7), anti-CD43 (clone, S7), FITC-conjugated anti-Ly6G (clone, 1A8), anti-IgD (clone, 11–26 C2a), IL-4 (clone, 11B11), anti-PD-1 (clone, 29 F.1A12), PerCP Cy5.5-conjugated anti-Ly6C (clone, AL-21), -CD45.1 (clone, A20), anti-IL-17 (clone, TC11-18H10), anti-mouse alpha muscle actin (Abcam, ab8211-500), Allophycocyanin (APC)-conjugated anti-CD11c (clone, HL3), anti-CD5 (clone, 53–7.3), BV421-conjugated anti-CD11b (clone, M1/70), anti-CD62L (clone, MEL-14), anti-IgG1 (clone, A110-1), AlexaFluor 700-conjugated anti-CD3ε (clone, 145–2 C11) -anti-IFN-γ (clone, XMG1.2), BV510-anti-CD4 (clone, RM4-5) and anti-B220 (clone, RA3-6B2), APC-Cy7-conjugated anti-CD19 (clone, 1D3) and anti-CD8 (clone, 53–6.7), BV786-conjugated anti-IgE (clone, R35-72) and anti-IL-33R (ST2) (clone, U29-93), biotin-conjugated anti-IgM (clone, AF-78), anti-CD95 (clone, Jo2), anti-CD249 (clone, BP-1), anti-CD45 (clone, 30-F11) were purchased from BD Pharmingen (San Diego, CA). PE-Cyanine7 anti-F4/80 (clone, BM8), anti-IL-13 (clone, eBio13A), anti-CXCR5 (clone, L138D7), Alexa Fluor 700-conjugated anti-MHC II (clone, M5/114), Rabbit anti-BAIAP2L1 (Abcam, PA554000), Live/dead Fixable Yellow stain (Qdot605 dead cell exclusion dye) were purchased from eBiosciences. Biotin-labelled antibodies were detected by Texas Red-conjugated PE (BD Biosciences). PE-Goat anti-Rabbit IgG (Abcam, ab72465) was used to detect Rabbit anti-BAIAP2L1. For staining, cells ($1 \times 10^6$) were stained and washed in PBS, 3% FCS FACS buffer. For intracellular cytokine staining, cells were restimulated with phorbol myristate acetate (Sigma-Aldrich) (50 ng/mL), ionomycin (Sigma-Aldrich) (250 ng/mL), and monensin (Sigma-Aldrich) (200 mM in IMDM/10% FCS) for 5 hr at 37 °C then fixed in 2% PFA, permeabilised with Foxp3 transcriptional factor staining buffer kit (eBioscience) before intracellular staining with appropriate cytokine antibodies and acquisition through LSR Fortessa machine (BD Immunocytometry system, San Jose, CA, USA) and data was analysed using Flowjo software (Treestar, Ashland, OR, USA).

## Histology

Left upper lung lobes was fixed in 4% formaldehyde/PBS and embedded in paraffin. Tissue sections were stained with periodic acid-Schiff for mucus secretion, and haematoxylin and eosin (H&E) stain for inflammation. Slides were scanned at 20 x magnification on the virtual slide VS120 microscope (Olympus, Japan). Downstream processing of images was done through Image J (FIJI) for image extraction at series 15 and Ilastik software was used for mucus area quantification on whole lung sections. The data shown are representative of 1 experiment of 3 independent experiments (n=5–7 mice per experiment).

## Antibody and cytokine ELISAs

Antibody ELISAs were carried out as previously described (*Kirstein et al., 2016*) using 10 μg/ml HDM to coat for specific IgGs. Total IgE in serum was measured using anti-mouse IgE (BD Biosciences, 553413) to coat, mouse IgE ($\kappa$, anti-TNP, BD Biosciences, 557079) as standard and biotin anti-mouse IgE (BD Biosciences, 553419) as a secondary antibody.

For in vitro cytokine production analysis, single-cell suspensions were prepared from mediastinal lymph nodes of HDM-treated and littermate control mice. Cells (2×10⁵ cells, in 200 μL) were incubated for 5 days in RPMI/10% FCS (Delta Bioproducts, Kempton Park, South Africa) in 96-well plates. Cells were either stimulated with HDM (30 μg/mL) or plate-bound anti-CD3 (10 μg/mL) and supernatants were collected after a 5 day incubation period. Concentrations of IL-4, IL-5 (BD Biosciences), and IL-13 (R&D Systems, Minneapolis, Minn) were measured using ELISA assays according to the manufacturer's protocol.

## RNA extraction
Small lung lobe was frozen in Qiazol (Qiagen, Germany) and stored at −80 °C. Total RNA was isolated from the lysate using miRNeasy Mini kit (Qiagen, Germany) according to the manufacturer's instructions. RNA quantity and purity were measured using the ND-1000 NanoDrop spectrophotometer (Thermo Scientific, DE, USA).

## cDNA synthesis and RT-qPCR
For *Muc5a* gene expression analysis, 100 ng total RNA was reverse transcribed into cDNA using Transcriptor First Strand cDNA Synthesis Kit (Roche, Germany) according to the manufacturer's instructions. Quantitative real-time PCR (RT-qPCR) was performed using LightCycler 480 SYBR Green I Master (Roche, Germany) and *Muc5a* primers (IDT, CA, USA). Fold change in gene expression was calculated by the ΔΔCt method and normalized to *Actb* which was used as an internal control.

## Whole lung RNA sequencing
Whole lung RNA was extracted using RNAeasy kit (Qiagen, Germany) according to the manufacturer's instructions. We used an the Agilent 2100 Bioanalyzer (Agilent RNA 6000 Nano Kit) to do the total RNA sample QC: RNA concentration, RIN value, 28S/18 S and the fragment length distribution. The first step in the workflow involved purifying the poly-A-containing mRNA molecules using poly-T oligo-attached magnetic beads. Following purification, the mRNA was fragmented into small pieces using divalent cations under elevated temperature. The cleaved RNA fragments were copied into first strand cDNA using reverse transcriptase and random primers. This was followed by second strand cDNA synthesis using DNA Polymerase I and RNase H. These cDNA fragments had the addition of a single 'A' base and subsequent ligation of the adapter. The products were then purified and enriched with PCR amplification. PCR yields were quantified by Qubit and pooled samples together to make a single-strand DNA circle (ssDNA circle), which gave the final library. DNA nanoballs (DNBs) were generated with the ssDNA circle by rolling circle replication (RCR) to enlarge the fluorescent signals at the sequencing process. The DNBs were loaded into the patterned nanoarrays and pair-end reads of 100 bp were read through on the DNBseq platform for the following data analysis study. For this step, the DNBseq platform combines the DNA nanoball-based nanoarrays and stepwise sequencing using Combinational Probe-Anchor Synthesis Sequencing Method.

## Bioinformatics workflow
The ribosomal RNA (rRNA) was first removed using SortMeRNA. We then did the Fastq file quality control using FastQC and MultiQC software to assess the quality of the raw reads, followed by adapter trimming using Trim Galore. The reads were then aligned to the mouse reference genome (mm10_UCSC_20180903) using STAR aligner. The map read counts was then extracted using featureCounts. The gene differential analysis was conducted using DESeq2. The genes with LFC ≥ 2 and adjusted *p*-value ≤ 0.05 was used to do the gene ontology over-representation analysis done using the clusterProfiler (*Wu et al., 2021*) Bioconductor package. The Benjamini-Hochberg method was used for multiple test correction.

## Human bronchial airway smooth muscle cell culture
A healthy human bronchial smooth muscle cell line (BSMC) was obtained from Lonza (Catalog #: CC-2576) and cultured in Smooth Muscle Growth Medium (Lonza) supplemented with Insulin (CC-4021D), human Fibroblastic Growth Factor-B (CC-4068D), Gentamicin sulfate-Amphotericin-1000 (CC-4081D), 5% Foetal Bovine Serum (CC-4102D), human Epidermal Growth Factor (CC-4230D) (all purchased from Lonza) in a T25 flask (Lonza) until confluence. After two passages in the T75 flask, confluent cells were seeded at 1.6×10⁵ BSMC cells onto 24-well trays and immediately transfected

with CRISPR/Cas9 single guide RNAs. No mycoplasma contamination was detected during the time of culturing. We verified human BSMCs by staining with alpha smooth muscle antibody.

## CRISPR/Cas9 single guide RNA transfections

Single guide RNAs targeting human *BAIAP2L1* (# CD.Cas9.CXVQ6494.AA); *ERDR1*, (# CD.Cas9. YFVV2490.AA); HPRT Negative Control (#Alt-R CRISPR-Cas9crRNA) were purchased (IDT, CA, USA via WhiteScientific PTY LTD). Single guide RNA (1 µM) and Cas9 enzyme (1 µM) in Opti-MEM medium (Life Technologies, Carlsbad, CA, USA) were transfected using Lipofectamine RNAi Max 1000 (Thermo Scientific) into BSMC cells ($1.6×10^5$) per well and either stimulated with Acetylcholine (10 µM), recombinant human IL-13 (100 ng/mL), recombinant human TNF-α (10 ng/mL) or left unstimulated for 48 hr at 37 °C, 5% $CO_2$ incubator. Gene deletion was confirmed by DNA extraction (Wizard Genomic DNA Purif. Kit, Promega) and PCR amplification of target genes using *BAIAP2L1* forward (GTCC CGGGGGGCCCGA) and reverse (AAGCGCCCAAGAATGTGGGG) primers, and product run on a 1.2% agarose gel.

## Cytotoxic detection assay

To measure cytotoxicity after single guide RNA transfection, lactate dehydrogenase (LDH) was measured in supernatants collected at 48 hr post-transfection using cytotoxicity detection kit [PLUS] Assay (CYTODET-RO, Roche) according to the manufacturer's instructions and plates were read at 490 nm.

## Single-cell force cytometry using fluorescently labelled elastomeric contractible surface (FLECS) technology

BSMCs were seeded on Fibronectin-coated elastomeric micropatterns (Forcyte Biotechnologies #F2AX0G03 Y, 50 microns, Alexa Fluor 488-bound Fibrinogen, 24-well plate) at a concentration of 75,000 cells per well in Smooth muscle growth medium (SmGM) (Lonza) and left to adhere and spread for 90 min. Unattached cells were then removed and fresh SmGM medium supplemented with rhTNF-α (10 ng/mL)/rhIL-13 (100 ng/mL)/ACh (10 µM) was added. BSMCs were stimulated for 3 hr before being fixed in pre-warmed to 37 °C 4% PFA. Fixed samples were washed and then stained with ATTO-565 phalloidin (ATTO-TEC) and DAPI (Life Technologies) and imaged on a StellarVision microscope using Synthetic Aperture Optics technology (Optical Biosystems). All images were analyzed using FiJi software by measuring the length of each micropattern per condition after stimulation and subtracting the length of the unstimulated micropattern of the same condition.

## Immunofluorescent

Left upper lung lobes were fixed in 4% formaldehyde/PBS and embedded in paraffin. Tissue sections (5 µm) were deparaffinised and hydrated in water before antigen retrieval in boiling 10 mM citrate buffer (pH 6, pressure cooker for 2 min). Tissue was incubated with Rabbit anti-BAIAP2L1 (Abcam, PA554000) in PBS-Tween (1:250, overnight, 4 °C) and FITC anti-mouse alpha muscle actin (Abcam, ab8211-500, 1:100, overnight, 4 °C). Tissues were washed with PBS-Tween before incubation with secondary antibody PE-Goat anti-Rabbit IgG (Abcam, ab72465) in PBS-Tween (1:500, RT in the dark for 30 min). Tissues were washed with PBS-Tween before mounting in DAPI mounting media (Thermo Fisher). Fluorescent images were captured at 60 x magnification using LSM880 Airy Scan (Zeiss, UK).

## Western blotting

Protein harvest: RIPA lysis buffer supplemented with protease inhibitors was added to the lung tissue. The whole-cell lysates were incubated on ice for 20 min and vortexed every 5 min for 60 s. Then, the lysate was centrifuged at 3000 rpm for 15 min at 4 °C, and the supernatant was collected. Total protein concentrations were determined using the Pierce BCA Assay kit.

SDS-PAGE: 20 µg of each protein sample was mixed with Laemmli buffer and heated for 10 min at 100 °C. Equal amounts of protein samples were loaded into 5% acrylamide stacking gel and a 12% acrylamide separating gel, along with 5 µL PageRulerTM Plus Prestained protein ladder (catalogue #: 26619, Thermo ScientificTM), and separated by electrophoresis at 100 V. Proteins were transferred onto nitrocellulose membranes (Bio-Rad) using the Bio-Rad Mini Trans-Blot Cell according to the manufacturer's instructions.

After protein transfer, the nitrocellulose membranes were stained with Ponceau stain for approximately 1–3 min to visualize if the proteins had been transferred. Once the protein bands were identified, the membrane was washed with ddH$_2$O until 80% of the Ponceau staining is removed. The remaining 20% of the stain allowed the accurate segmentation of the membrane to separate high molecular weight proteins from low molecular weight proteins. The nitrocellulose membranes were blocked with 5% (w/v) non-fat milk, 0.01% Tween-20 in TBST on the shaker at room temperature for 1 hr. The low and high molecular weight membrane proteins were incubated with respective primary antibodies (Rabbit anti-human BAIAP2L1, PA5-54000, Invitrogen) or (Goat anti-mouse GAPDH, sc 365062, Santa Cruz) overnight shaking in the 4 °C cold room. Membranes were washed four times with 1x TBST in 15 min intervals. Horseradish peroxidase (HRP) secondary antibodies were added to their respective membranes and incubated for 1 hr at room temperature, with shaking. Subsequently, membranes were washed three times in 1xTBST in 15 min intervals. To visualise the protein bands on autoradiographic film (Santa Cruz Biotechnology), Lumino Glo substrate kit was added to each membrane. Briefly, the substrate luminol was oxidised by hydrogen peroxide in the presence of the catalyst HRP to yield a chemiluminescent product.

## Plating of faecal pellets to confirm a reduction in bacterial colonies following antibiotic treatment

The faecal pellet was collected from each individual mouse treated with antibiotics as well as control groups not treated with antibiotics. The faecal pellet was weighed and resuspended in sterile PBS at a concentration of 50 mg/mL. The sample was resuspended until homogenous, and 1/100 dilution was made in sterile PBS. 50 μL of 1/100 dilutions was plated on tryptic soy agar plates and spread evenly across the plate before incubating at 37 °C for 16 hr. Individual colonies were counted manually.

### Statistical analysis

P-values were calculated in GraphPad Prism 6 (GraphPad Software, Inc) by using nonparametric Mann-Whitney Student's t-test or two-way ANOVA with Bonferroni's post-test for multiple comparisons, and results are presented as standard error of the mean (SEM) or mean of standard deviation (SD). Differences were considered significant if p was <0.05.

## Acknowledgements

We thank the UCT Research Animal Facility for maintaining mice. Wendy Green and Munadia Ansari for genotyping mice. We are grateful to Lizette Fick, Matt Darby, and Raygaana Jacobs for their excellent histology services and the UCT Confocal Microscopy Core Facilities. We are grateful to Ronnie Dreyer for the excellent cell sorting and Flow Cytometry Core Facility. We thank FlowJo for providing free service to Africa. This work was supported by ICGEB, Cape Town Component, Medical Research Council (MRC) South Africa as well as support by the South African National Research Foundation (NRF) Research Chair initiative (SARChi) and Wellcome Trust CIDRI-Africa (203135Z/16/Z) to FB. SH is supported by NRF Thuthuka Grant (117721), NRF Competitive Support for Unrated Researcher (138072), MRC South Africa under Self-initiated grant. NM was supported by the South African MRC PhD Fellowship and ATAP Fellowship at UCT. AN was supported by NRF MSc Scholarship. This research was funded in whole, or in part, by the Wellcome Trust [Grant number 203135Z/16/Z]. For the purpose of open access, the author has applied a CC BY public copyright licence to any Author Accepted Manuscript version arising from this submission.

## Additional information

### Funding

| Funder | Grant reference number | Author |
| --- | --- | --- |
| South African Medical Research Council | | Frank Brombacher |

| Funder | Grant reference number | Author |
|---|---|---|
| National Research Foundation | Research Chair initiative (SARChi) | Frank Brombacher |
| Wellcome Trust | 203135Z/16/Z | Frank Brombacher |
| National Research Foundation | Thuthuka 117721 | Sabelo Hadebe |
| National Research Foundation | CSUR 138072 | Sabelo Hadebe |
| South African Medical Research Council | Self initiated grant | Sabelo Hadebe |
| South African Medical Research Council | PhD Fellowship | Nontobeko Mthembu |
| University of Cape Town | ATAP Fellowship | Nontobeko Mthembu |
| National Research Foundation | MSc Scholarship | Amkele Ngomti |

The funders had no role in study design, data collection and interpretation, or the decision to submit the work for publication. For the purpose of Open Access, the authors have applied a CC BY public copyright license to any Author Accepted Manuscript version arising from this submission.

## Author contributions

Sabelo Hadebe, Conceptualization, Resources, Data curation, Software, Formal analysis, Supervision, Funding acquisition, Validation, Investigation, Visualization, Methodology, Writing – original draft, Project administration, Writing – review and editing; Anca Flavia Savulescu, Data curation, Formal analysis, Supervision, Investigation, Visualization, Methodology, Writing – review and editing; Jermaine Khumalo, Data curation, Formal analysis, Visualization, Methodology, Writing – review and editing; Katelyn Jones, Data curation, Formal analysis, Validation, Visualization, Methodology, Writing – review and editing; Sandisiwe Mangali, Nontobeko Mthembu, Formal analysis, Investigation, Visualization, Methodology, Writing – review and editing; Fungai Musaigwa, Hlumani Ndlovu, Formal analysis, Visualization, Writing – review and editing; Welcome Maepa, Formal analysis, Validation, Writing – review and editing; Amkele Ngomti, Formal analysis, Investigation, Writing – review and editing; Martyna Scibiorek, Formal analysis, Writing – review and editing; Javan Okendo, Data curation, Software, Investigation, Methodology, Writing – review and editing; Frank Brombacher, Conceptualization, Supervision, Funding acquisition, Investigation, Writing – review and editing

## Author ORCIDs

Sabelo Hadebe https://orcid.org/0000-0002-6049-8395
Anca Flavia Savulescu https://orcid.org/0000-0002-3428-0157
Welcome Maepa https://orcid.org/0000-0003-2881-5167
Javan Okendo https://orcid.org/0000-0001-8218-5448

## Ethics

Mice were housed in independently ventilated cages under specific pathogen-free conditions at the University of Cape Town Animal Facility. All mice were used at eight to 10 weeks of age and animal procedures were performed according to the strict recommendation by the South African Veterinary Council and were approved by the University of Cape Town Animal Ethics Committee (Reference number 014/019, 018/013 and 022/014).

Reviewer #4 (Public review): https://doi.org/10.7554/eLife.90531.4.sa1
Author response https://doi.org/10.7554/eLife.90531.4.sa2

# Additional files

## Supplementary files

MDAR checklist

## Data availability

The data related to this paper can be found at NCBI Sequence Read Archive (SRA) number PRJNA1288527. Single cell force cytometry data can be found at figshare under DOIs https://doi.org/10.6084/m9.figshare.29832320 and https://doi.org/10.6084/m9.figshare.29832263.

The following datasets were generated:

| Author(s) | Year | Dataset title | Dataset URL | Database and Identifier |
|---|---|---|---|---|
| Hadebe S | 2025 | Immunoglobulin M regulates airway hyperresponsiveness independent of T helper 2 allergic inflammation | https://www.ncbi.nlm.nih.gov/bioproject/PRJNA1288527/ | NCBI BioProject, PRJNA1288527 |
| Hadebe S | 2025 | Single-cell force cytometry using Fluorescently Labelled Elastomeric Contractible Surface (FLECS) technology A | https://doi.org/10.6084/m9.figshare.29832263 | figshare, 10.6084/m9.figshare.29832263 |
| Hadebe S | 2025 | Single-cell force cytometry using Fluorescently Labelled Elastomeric Contractible Surface (FLECS) technology B | https://doi.org/10.6084/m9.figshare.29832320 | figshare, 10.6084/m9.figshare.29832320 |

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
