## [Editor Report · eLife Assessment]

This **important** study demonstrates a reduction in airway hyperresponsiveness (one of the mechanisms of allergic asthma) in the absence of IgM in a house dust mite-induced mouse model of allergic asthma. While this result suggests a new mechanistic role for IgM, the proposed new function is not as yet robustly supported by the current experiments and thus the evidence remains **incomplete**. A connection between the findings and human disease is not established so far, but the study will be interest to clinical immunologists.

---

## [Referee Report · Reviewer #4 (Public review)]

Summary:

The authors sought to determine the role of IgM in a house dust mite (HDM)-induced Th2 allergic model. Specifically, they examined the effect of IgM deficiency by comparing airway hyperresponsiveness (AHR) and Th2 immune responses between wild-type (WT) and IgM knockout (KO) mice exposed to HDM. They found and reported a reduction in AHR among the KO mice. This finding was followed by experiments investigating the role of IgM in airway smooth muscle (ASM) contraction using a human cell line, based on two genes that were reportedly differentially expressed between lung tissues from WT and IgM KO mice following HDM exposure.

Strengths:

Knocking out IgM produced a clear phenotype of reduced airway hyperresponsiveness (AHR), suggesting a previously unreported role for IgM in this process. The authors conducted extensive experiments to elucidate this novel role of IgM.

Weaknesses:

Although a few differentially expressed genes (DEGs) are reported between WT HDM vs. IgM KO HDM and WT PBS vs. IgM KO PBS, the principal component analysis (PCA) did not show any group-specific clustering based on these DEGs. This undermines the strength of the authors' reliance on these results as the foundation for subsequent experiments.

Furthermore, if IgM does indeed have a demonstrable effect on airway smooth muscle (ASM), this could be more convincingly shown using in vitro muscle contraction assays with alternative methods.

---

## [Author Response]

The following is the authors’ response to the previous reviews

**Reviewer #1 (Public Review):**
Summary:The authors of this study sought to define a role for IgM in responses to house dust mites in the lung.Strengths:Unexpected observation about IgM biologyCombination of experiments to elucidate functionWeaknesses:Would love more connection to human disease

We thank the reviewer for these comments. At the time of this publication, we have not made a concrete link with human disease. While there is some anecdotal evidence of diseases such as Autoimmune glomerulonephritis, Hashimoto’s thyroiditis, Bronchial polyp, SLE, Celiac disease and other diseases in people with low IgM. Allergic disorders are also common in people with IgM deficiency, other studies have reported as high as 33-47%. The mechanisms for the high incidence of allergic diseases are unclear as generally, these patients have normal IgG and IgE levels. IgM deficiency may represent a heterogeneous spectrum of genetic defects, which might explain the heterogeneous nature of disease presentations.

**Reviewer #2 (Public Review):**
Summary:The manuscript by Hadebe and colleagues describes a striking reduction in airway hyperresponsiveness in Igm-deficient mice in response to HDM, OVA and papain across the B6 and BALB-c backgrounds. The authors suggest that the deficit is not due to improper type 2 immune responses, nor an aberrant B cell response, despite a lack of class switching in these mice. Through RNA-Seq approaches, the authors identify few differences between the lungs of WT and Igm-deficient mice, but see that two genes involved in actin regulation are greatly reduced in IgM-deficient mice. The authors target these genes by CRISPR-Cas9 in in vitro assays of smooth muscle cells to show that these may regulate cell contraction. While the study is conceptually interesting, there are a number of limitations, which stop us from drawing meaningful conclusions.Strengths:Fig. 1. The authors clearly show that IgMKO mice have striking reduced AHR in the HDM model, despite the presence of a good cellular B cell response.Weaknesses:Fig. 2. The authors characterize the cd4 t cell response to HDM in IGMKO mice.They have restimulated medLN cells with antiCD3 for 5 days to look for IL-4 and IL-13, and find no discernible difference between WT and KO mice. The absence of PBStreated WT and KO mice in this analysis means it is unclear if HDM-challenged mice are showing IL-4 or IL-13 levels above that seen at baseline in this assay.

We thank the Reviewer for this comment. We would like to mention that a very minimal level of IL-4 and IL-13 in PBS mice was detected. We have indicated with a dotted line on the Figure 2B to show levels in unstimulated or naïve cytokines. Please see Author response image 1 below from anti-CD3 stimulated cytokine ELISA data. The levels of these cytokines are very low (not detectable) and are not changed in control WT and IgM- KO mice challenge with PBS, this is also true for PMA/ionomycin-stimulated cells

**Author response image 1. sa2fig1:** 

The choice of 5 days is strange, given that the response the authors want to see is in already primed cells. A 1-2 day assay would have been better.

We agree with the reviewer that a shorter stimulation period would work. Over the years we have settled for 5-day re-stimulation for both anti-CD3 and HDM. We have tried other time points, but we consistently get better secretion of cytokines after 5 days.

It is concerning that the authors state that HDM restimulation did not induce cytokine production from medLN cells, since countless studies have shown that restimulation of medLN would induce IL-13, IL-5 and IL-10 production from medLN. This indicates that the sensitization and challenge model used by the authors is not working as it should.

We thank the reviewer for this observation. In our recent paper showing how antigen load affects B cell function, we used very low levels of HDM to sensitise and challenge mice (1 ug and 3 ug respectively). See below article, Hadebe et al., 2021 JACI. This is because Labs that have used these low HDM levels also suggested that antigen load impacts B cell function, especially in their role in germinal centres. We believe the reason we see low or undetectable levels of cytokines is because of this low antigen load sensitisation and challenge. In other manuscripts we have published or about to publish, we have shown that normal HDM sensitisation load (1 ug or 100 ug) and challenge (10 ug) do induce cytokine release upon restimulation with HDM. See the below article by Khumalo et al, 2020 JCI Insight (Figure 4A).

Sabelo Hadebe*, Jermaine Khumalo, Sandisiwe Mangali, Nontobeko Mthembu, Hlumani Ndlovu, Amkele Ngomti, Martyna Scibiorek, Frank Kirstein, Frank Brombacher*. Deletion of IL-4Ra signalling on B cells limits hyperresponsiveness depending on antigen load. doi.org/10.1016/j.jaci.2020.12.635.

Jermaine Khumalo, Frank Kirstein, Sabelo Hadebe*, Frank Brombacher*. IL-4Rα signalling in regulatory T cells is required for dampening allergic airway inflammation through inhibition of IL-33 by type 2 innate lymphoid cells. JCI Insight. 2020 Oct 15;5(20):e136206. doi: 10.1172/jci.insight.136206

The IL-13 staining shown in panel c is also not definitive. One should be able to optimize their assays to achieve a better level of staining, to my mind.

We agree with the reviewer that much higher IL-13-producing CD4 T cells should be observed. We don’t think this is a technical glitch or non-optimal set-up as we see much higher levels of IL-13-producing CD4 T cells when using higher doses of HDM to sensitise and challenge, say between 7 -20% in WT mice (see Author response image 2 of lung stimulated with PMA/ionomycin+Monensin, please note this is for illustration purposes only and it not linked to the current manuscript, its merely to demonstrate a point from other experiments we have conducted in the lab).

In d-f, the authors perform a serum transfer, but they only do this once. The half life of IgM is quite short. The authors should perform multiple naïve serum transfers to see if this is enough to induce FULL AHR.

We thank the reviewer for this comment. We apologise if this was not clear enough on the Figure legend and method, we did transfer serum 3x, a day before sensitisation, on the day of sensitisation and a day before the challenge to circumvent the short life of IgM. In our subsequent experiments, we have now used busulfan to deplete all bone marrow in IgM-deficient mice and replace it with WT bone marrow and this method restores AHR (Figure 3B).

This now appears in line 515 to 519 and reads

Adoptive transfer of naïve serum

Naïve wild-type mice were euthanised and blood was collected via cardiac puncture before being spun down (5500rpm, 10min, RT) to collect serum. Serum (200µL) was injected intraperitoneally into IgM-deficient mice. Serum was injected intraperitoneally at day -1, 0, and a day before the challenge with HDM (day 10).

The presence of negative values of total IgE in panel F would indicate some errors in calculation of serum IgE concentrations.

We thank the reviewer for this observation. For better clarity, we have now indicated these values as undetected in Figure 2F, as they were below our detection limit.

Overall, it is hard to be convinced that IgM-deficiency does not lead to a reduction in Th2 inflammation, since the assays appear suboptimal.

We disagree with the reviewer in this instance, because we have shown in 3 different models and in 2 different strains and 2 doses of HDM (high and low) that no matter what you do, Th2 remains intact. Our reason for choosing low dose HDM was based on our previous work and that of others, which showed that depending on antigen load, B cells can either be redundant or have functional roles. Since our interest was to tease out the role of B cells and specifically IgM, it was important that we look at a scenario where B cells are known to have a function (low antigen load). We did find similar findings at high dose of HDM load, but effects on AHR were not as strong, but Th2 was not changed, in fact in some instances Th2 was higher in IgM-deficient mice.

Fig. 3. Gene expression differences between WT and KO mice in PBS and HDM challenged settings are shown. PCA analysis does not show clear differences between all four groups, but genes are certainly up and downregulated, in particular when comparing PBS to HDM challenged mice. In both PBS and HDM challenged settings, three genes stand out as being upregulated in WT v KO mice. these are Baiap2l1, erdr1 and Chil1.

Noted

Fig. 4. The authors attempt to quantify BAIAP2L1 in mouse lungs. It is difficult to know if the antibody used really detects the correct protein. A BAIAP2L1-KO is not used as a control for staining, and I am not sure if competitive assays for BAIAP2L1 can be set up. The flow data is not convincing. The immunohistochemistry shows BAIAP2L1 (in red) in many, many cells, essentially throughout the section. There is also no discernible difference between WT and KO mice, which one might have expected based on the RNA-Seq data. So, from my perspective, it is hard to say if/where this protein is located, and whether there truly exists a difference in expression between wt and ko mice.We thank the reviewer for this comment. We are certain that the antibody does detect BAIAP2L1, we have used it in 3 assays, which we admit may show varying specificities since it’s a Polyclonal antibody. However, in our western blot (Figure 5A), the antibody detects a band at 56.7kDa, apart from what we think are isoforms. We agree that BAIAP2L1 is expressed by many cell types, including CD45+ cells and alpha smooth muscle negative cells and we show this in our Figure 5 – figure supplement 1A and B. Where we think there is a difference in expression between WT and IgM-deficient mice is in alpha-smooth muscle-positive cells. We have tested antibodies from different companies (Proteintech and Abcam), and we find similar findings. We do not have access to BAIAP2L1 KO mice and to test specificity, we have also used single stain controls with or without secondary antibody and isotype control which show no binding in western blot and Immunofluorescence assays and Fluorescence minus one antibody in Flow cytometry, so that way we are convinced that the signal we are seeing is specific to BAIAP2L1.

Here we have also added additional Flow cytometry images using anti-BAIAP2L1 (clone 25692-1-AP) from Proteintech

**Author response image 3. sa2fig3:** 

Figure similar to Figure 5C and Figure 5 -figure supplement 1A and B.Fig. 5 and 6. The authors use a single cell contractility assay to measure whether BAIAP2L1 and ERDR1 impact on bronchial smooth muscle cell contractility. I am not familiar with the assay, but it looks like an interesting way of analysing contractility at the single cell level.The authors state that targeting these two genes with Cas9gRNA reduces smooth muscle cell contractility, and the data presented for contractility supports this observation. However, the efficiency of Cas9-mediated deletion is very unclear. The authors present a PCR in supp fig 9c as evidence of gene deletion, but it is entirely unclear with what efficiency the gene has been deleted. One should use sequencing to confirm deletion. Moreover, if the antibody was truly working, one should be able to use the antibody used in Fig 4 to detect BAIAP2L1 levels in these cells. The authors do not appear to have tried this.

We thank the reviewer for these observations. We are in a process to optimise this using new polyclonal BAIAP2L1 antibodies from other companies, since the one we have tried doesn’t seem to work well on human cells via western blot. So hopefully in our new version, we will be able to demonstrate this by immunofluorescence or western blot.

Other impressions:The paper is lacking a link between the deficiency of IgM and the effects on smooth muscle cell contraction.The levels of IL-13 and TNF in lavage of WT and IGMKO mice could be analysed.

We have measured Th2 cytokine IL-13 in BAL fluid and found no differences between IgM-deficient mice and WT mice challenged with HDM (Author response image 4 below). We could not detected TNF-alpha in the BAL fluid, it was below detection limit.

Figure legend. IL-13 levels are not changed in IgM-deficient mice in the lung. Bronchoalveolar lavage fluid in WT or IgM-deficient mice sensitised and challenged with HDM. TNF-a levels were below the detection limit.

**Author response image 4. sa2fig4:** 

Moreover, what is the impact of IgM itself on smooth muscle cells? In the Fig. 7 schematic, are the authors proposing a direct role for IgM on smooth muscle cells? Does IgM in cell culture media induce contraction of SMC? This could be tested and would be interesting, to my mind.

We thank the Reviewer for these comments. We are still trying to test this, unfortunately, we have experienced delays in getting reagents such as human IgM to South Africa. We hope that we will be able to add this in our subsequent versions of the article. We agree it is an interesting experiment to do even if not for this manuscript but for our general understanding of this interaction at least in an in vitro system.

**Reviewer #3 (Public Review):**
Summary:This paper by Sabelo et al. describes a new pathway by which lack of IgM in the mouse lowers bronchial hyperresponsiveness (BHR) in response to metacholine in several mouse models of allergic airway inflammation in Balb/c mice and C57/Bl6 mice. Strikingly, loss of IgM does not lead to less eosinophilic airway inflammation, Th2 cytokine production or mucus metaplasia, but to a selective loss of BHR. This occurs irrespective of the dose of allergen used. This was important to address since several prior models of HDM allergy have shown that the contribution of B cells to airway inflammation and BHR is dose dependent.After a description of the phenotype, the authors try to elucidate the mechanisms. There is no loss of B cells in these mice. However, there is a lack of class switching to IgE and IgG1, with a concomitant increase in IgD. Restoring immunoglobulins with transfer of naïve serum in IgM deficient mice leads to restoration of allergen-specific IgE and IgG1 responses, which is not really explained in the paper how this might work. There is also no restoration of IgM responses, and concomitantly, the phenotype of reduced BHR still holds when serum is given, leading authors to conclude that the mechanism is IgE and IgG1 independent. Wild type B cell transfer also does not restore IgM responses, due to lack of engraftment of the B cells. Next authors do whole lung RNA sequencing and pinpoint reduced BAIAP2L1 mRNA as the culprit of the phenotype of IgM-/- mice. However, this cannot be validated fully on protein levels and immunohistology since differences between WT and IgM KO are not statistically significant, and B cell and IgM restoration are impossible. The histology and flow cytometry seems to suggest that expression is mainly found in alpha smooth muscle positive cells, which could still be smooth muscle cells or myofibroblasts. Next therefore, the authors move to CRISPR knock down of BAIAP2L1 in a human smooth muscle cell line, and show that loss leads to less contraction of these cells in vitro in a microscopic FLECS assay, in which smooth muscle cells bind to elastomeric contractible surfaces.Strengths:(1) There is a strong reduction in BHR in IgM-deficient mice, without alterations in B cell number, disconnected from effects on eosinophilia or Th2 cytokine production.(2) BAIAP2L1 has never been linked to asthma in mice or humansWeaknesses:(1) While the observations of reduced BHR in IgM deficient mice are strong, there is insufficient mechanistic underpinning on how loss of IgM could lead to reduced expression of BAIAP2L1. Since it is impossible to restore IgM levels by either serum or B cell transfer and since protein levels of BAIAP2L1 are not significantly reduced, there is a lack of a causal relationship that this is the explanation for the lack of BHR in IgMdeficient mice. The reader is unclear if there is a fundamental (maybe developmental) difference in non-hematopoietic cells in these IgM-deficient mice (which might have accumulated another genetic mutation over the years). In this regard, it would be important to know if littermates were newly generated, or historically bred along with the KO line.

We thank the reviewer for asking this question and getting us to think of this in a different way. This prompted us to use a different method to try and restore IgM function and since our animal facility no longer allows irradiation, we opted for busulfan. We present this data as new data in Figure 3. We had to go back and breed this strain and then generated bone marrow chimeras. What we have shown now with chimeras is that if we can deplete bone marrow from IgM-deficient mice and replace it with congenic WT bone marrow when we allow these mice to rest for 2 months before challenge with HDM (Figure 3 -figure supplement 1A-C) We also show that AHR (resistance and elastance) is partially restored in this way (Figure 3A and B) as mice that receive congenic WT bone marrow after chemical irradiation can mount AHR and those that receive IgM-deficient bone marrow, can’t mount AHR upon challenge with HDM. If the mice had accumulated an unknown genetic mutation in non-hematopoietic cells, the transfer of WT bone marrow would not make a difference. So, we don’t believe the colony could have gained a mutation that we are unaware of. We have also shipped these mice to other groups and in their hands, this strains still only behaves as an IgM only knockout mice. See their publication below.

Mark Noviski, James L Mueller, Anne Satterthwaite, Lee Ann Garrett-Sinha, Frank Brombacher, Julie Zikherman 2018. IgM and IgD B cell receptors differentially respond to endogenous antigens and control B cell fate. eLife 2018;7:e35074. DOI: https://doi.org/10.7554/eLife.35074

we have also added methods for bone marrow chimaeras and added results sections and new Figures related to these methods.

Methods appear in line 521-532 of the untracked version of the article.

Busulfan Bone marrow chimeras

WT (CD45.2) and IgM^-/-^ (CD45.2) congenic mice were treated with 25 mg/kg busulfan (Sigma-Aldrich, Aston Manor, South Africa) per day for 3 consecutive days (75 mg/kg in total) dissolved in 10% DMSO and Phosphate buffered saline (0.2mL, intraperitoneally) to ablate bone marrow cells. Twenty-four hours after last administration of busulfan, mice were injected intravenously with fresh bone marrow (10x10^6^ cells, 100µL) isolated from hind leg femurs of either WT (CD45.1) or IgM^-/-^ mice [33]. Animals were then allowed to complement their haematopoietic cells for 8 weeks. In some experiments the level of bone marrow ablation was assessed 4 days post-busulfan treatment in mice that did not receive donor cells. At the end of experiment level of complemented cells were also assessed in WT and IgM^-/-^ mice that received WT (CD45.1) bone marrow.

Results appear in line 198-228 of the untracked version of the article

Replacement of IgM-deficient mice with functional hematopoietic cells in busulfan mice chimeric mice restores airway hyperresponsiveness.

We then generated bone marrow chimeras by chemical radiation using busulfan (Montecino-Rodriguez and Dorshkind, 2020). We treated mice three times with busulfan for 3 consecutive days and after 24 hrs transferred naïve bone marrow from congenic CD45.1 WT mice or CD45.2 IgM KO mice (Figure 3A and Figure 3 -figure supplement 1A). We showed that recipient mice that did not receive donor bone marrow after 4 days post-treatment had significantly reduced lineage markers (CD45^+^Sca-1^+^) or lineage negative (Lin^-^) cells in the bone marrow when compared to untreated or vehicle (10% DMSO) treated mice (Figure 3 -figure supplements 1B-C). We allowed mice to reconstitute bone marrow for 8 weeks before sensitisation and challenge with low dose HDM (Figure 3A). We showed that WT (CD45.2) recipient mice that received WT (CD45.1) donor bone marrow had higher airway resistance and elastance and this was comparable to IgM KO (CD45.2) recipient mice that received donor WT (CD45.1) bone marrow (Figure 3B). As expected, IgM KO (CD45.2) recipient mice that received donor IgM KO (CD45.2) bone marrow had significantly lower AHR compared to WT (CD45.2) or IgM KO (CD45.2) recipient mice that received WT (CD45.1) bone marrow (Figure 3B). We confirmed that the differences observed were not due to differences in bone marrow reconstitution as we saw similar frequencies of CD45.1 cells within the lymphocyte populations in the lungs and other tissues (Figure 3 -figure supplement 1D). We observed no significant changes in the lung neutrophils, eosinophils, inflammatory macrophages, CD4 T cells or B cells in WT or IgM KO (CD45.2) recipient mice that received donor WT (CD45.1/CD45.2) or IgM KO (CD45.2) bone marrow when sensitised and challenged with low dose HDM (Figure 3C).

Restoring IgM function through adoptive reconstitution with congenic CD45.1 bone marrow in non-chemically irradiated recipient mice or sorted B cells into IgM KO mice (Figure 2 -figure supplement 1A) did not replenish IgM B cells to levels observed in WT mice and as a result did not restore AHR, total IgE and IgM in these mice (Figure 2 -figure supplements 1B-C).

The 2 new figures are Figure 3 which moved the rest of the Figures down and Figure 3- figure supplement 1AD, which also moved the rest of the supplementary figures down.

Discussion appears in line 410-419 of the untracked version of the article.To resolve other endogenous factors that could have potentially influenced reduced AHR in IgM-deficient mice, we resorted to busulfan chemical irradiation to deplete bone marrow cells in IgM-deficient mice and replace bone marrow with WT bone marrow. While it is well accepted that busulfan chemical irradiation partially depletes bone marrow cells, in our case it was not possible to pursue other irradiation methods due to changes in ethical regulations and that fact that mice are slow to recover after gamma rays irradiation. Busulfan chemical irradiation allowed us to show that we could mostly restore AHR in IgM-deficient recipient mice that received donor WT bone marrow when challenged with low dose HDM.

(2) There is no mention of the potential role of complement in activation of AHR, which might be altered in IgM-deficient mice

We thank the reviewer for this comment. We have not directly looked at complement in this instance, however, from our previous work on C3 knockout mice, there have been comparable AHR to WT mice under the HDM challenge.

(3) What is the contribution of elevated IgD in the phenotype of the IgM-deficient mice. It has been described by this group that IgD levels are clearly elevated

We thank the reviewer for this question. We believe that IgD is essentially what drives partial class switching to IgG, we certainly have shown that in the case of VSV virus and Trypanosoma congolense and *Trypanosoma brucei* brucei that elevated IgD drive delayed but effective IgG in the absence of IgM (Lutz et al, 2001, Nature). This is also confirmed by Noviski et al., 2018 eLife study where they show that both IgM and IgD do share some endogenous antigens, so its likely that external antigens can activate IgD in a similar manner to prompt class switching.

(4) How can transfer of naïve serum in class switching deficient IgM KO mice lead to restoration of allergen specific IgE and IgG1?

We thank the Reviewer for these comments, we believe that naïve sera transferred to IgM deficient mice is able to bind to the surface of B cells via IgM receptors (FcμR / Fcα/μR), which are still present on B cells and this is sufficient to facilitate class switching. Our IgM KO mouse lacks both membrane-bound and secreted IgM, and transferred serum contains at least secreted IgM which can bind to surfaces via its Fc portion. We measured HDM-specific IgE and we found very low levels, but these were not different between WT and IgM KO adoptively transferred with WT serum. We also detected HDM-specific IgG1 in IgM KO transferred with WT sera to the same level as WT, confirming a possible class switching, of course, we can’t rule out that transferred sera also contains some IgG1. We also can’t rule out that elevated IgD levels can partially be responsible for class switched IgG1 as discussed above.

In the discussion line 463-464, we also added the following

“We speculate that IgM can directly activate smooth muscle cells by binding a number of its surface receptors including FcμR, Fcα/μR and pIgR (Liu et al., 2019; Nguyen et al., 2017b; Shibuya et al., 2000). IgM binds to FcμR strictly, but shares Fcα/μR and pIgR with IgA (Liu et al., 2019; Michaud et al., 2020; Nguyen et al., 2017b). Both Fcα/μR and pIgR can be expressed by non-structural cells at mucosal sites (Kim et al., 2014; Liu et al., 2019). We would not rule out that the mechanisms of muscle contraction might be through one of these IgM receptors, especially the ones expressed on smooth muscle cells(Kim et al., 2014; Liu et al., 2019). Certainly, our future studies will be directed towards characterizing the mechanism by which IgM potentially activates the smooth muscle.”

We have discussed this section under Discussion section, line 731 to 757. In addition, since we have now performed bone marrow chimaeras we have further added the following in our discussion in line 410-419.

To resolve other endogenous factors that could have potentially influenced reduced AHR in IgM-deficient mice, we resorted to busulfan chemical irradiation to deplete bone marrow cells in IgM-deficient mice and replace bone marrow with WT bone marrow. While it is well accepted that busulfan chemical irradiation partially depletes bone marrow cells, in our case it was not possible to pursue other irradiation methods due to changes in ethical regulations and that fact that mice are slow to recover after gamma rays irradiation. Busulfan chemical irradiation allowed us to show that we could mostly restore AHR in IgM-deficient recipient mice that received donor WT bone marrow when challenged with low dose HDM.

We removed the following lines, after performing bone marrow chimaeras since this changed some aspects.

Our efforts to adoptively transfer wild-type bone marrow or sorted B cells into IgMdeficient mice were also largely unsuccessful partly due to poor engraftment of wildtype B cells into secondary lymphoid tissues. Natural secreted IgM is mainly produced by B1 cells in the peritoneal cavity, and it is likely that any transfer of B cells via bone marrow transfer would not be sufficient to restore soluble levels of IgM^3,10^.

(5) lpha smooth muscle antigen is also expressed by myofibroblasts. This is insufficiently worked out. The histology mentions "expression in cells in close contact with smooth muscle". This needs more detail since it is a very vague term. Is it in smooth muscle or in myofibroblasts.

We appreciate that alpha-smooth muscle actin-positive cells are a small fraction in the lung and even within CD45 negative cells, but their contribution to airway hyperresponsiveness is major. We also concede that by immunofluorescence BAIAP2L1 seems to be expressed by cells adjacent to alpha-smooth muscle actin (Figure 5B), however, we know that cells close to smooth muscle (such as extracellular matrix and myofibroblasts) contribute to its hypertrophy in allergic asthma.

James AL, Elliot JG, Jones RL, Carroll ML, Mauad T, Bai TR, et al. Airway Smooth Muscle Hypertrophy and Hyperplasia in Asthma. Am J Respir Crit Care Med [Internet]. 2012; 185:1058–64. Available from: https://doi.org/10.1164/rccm.201110-1849OC

(6) Have polymorphisms in BAIAP2L1 ever been linked to human asthma?

No, we have looked in asthma GWAS studies, at least summary statistics and we have not seen any SNPs that could be associated with human asthma.

(7) IgM deficient patients are at increased risk for asthma. This paper suggests the opposite. So the translational potential is unclear

We thank the reviewer for these comments. At the time of this publication, we have not made a concrete link with human disease. While there is some anecdotal evidence of diseases such as Autoimmune glomerulonephritis, Hashimoto’s thyroiditis, Bronchial polyp, SLE, Celiac disease and other diseases in people with low IgM. Allergic disorders are also common in people with IgM deficiency as the reviewer correctly points out, other studies have reported as high as 33-47%. The mechanisms for the high incidence of allergic diseases are unclear as generally, these patients have normal or higher IgG and IgE levels. IgM deficiency may represent a heterogeneous spectrum of genetic defects, which might explain the heterogeneous nature of disease presentations.